# *Legionella* para-effectors target chromatin and promote bacterial replication

Daniel Schator [1,2,5], Sonia Mondino[1,6], Jérémy Berthelet[3,7], Cristina Di Silvestre[1], Mathilde Ben Assaya[4], Christophe Rusniok[1], Fernando Rodrigues-Lima [3], Annemarie Wehenkel [4], Carmen Buchrieser [1] ✉ & Monica Rolando[1] ✉

*Legionella pneumophila* replicates intracellularly by secreting effectors via a type IV secretion system. One of these effectors is a eukaryotic methyltransferase (RomA) that methylates K14 of histone H3 (H3K14me3) to counteract host immune responses. However, it is not known how *L. pneumophila* infection catalyses H3K14 methylation as this residue is usually acetylated. Here we show that *L. pneumophila* secretes a eukaryotic-like histone deacetylase (LphD) that specifically targets H3K14ac and works in synergy with RomA. Both effectors target host chromatin and bind the HBO1 histone acetyltransferase complex that acetylates H3K14. Full activity of RomA is dependent on the presence of LphD as H3K14 methylation levels are significantly decreased in a *ΔlphD* mutant. The dependency of these two chromatin-modifying effectors on each other is further substantiated by mutational and virulence assays revealing that the presence of only one of these two effectors impairs intracellular replication, while a double knockout (*ΔlphDΔromA*) can restore intracellular replication. Uniquely, we present evidence for "para-effectors", an effector pair, that actively and coordinately modify host histones to hijack the host response. The identification of epigenetic marks modulated by pathogens has the potential to lead to the development of innovative therapeutic strategies to counteract bacterial infection and strengthening host defences.

The accessibility of chromatin to transcription factors and the subsequent changes in gene expression are key regulatory mechanism in eukaryotic cells. The process of changing this accessibility is known as chromatin remodeling. The dynamic modification of histones, the small basic proteins that DNA is wrapped around to form chromatin, is one of the most studied mechanisms of chromatin remodeling. The so-called histone tails – peptide sequences reaching out of the core histone structure – are subjected to a variety of post-translational modifications (PTMs)[1], such as acetylation, methylation, ubiquitination, or phosphorylation[2]. Methylation and acetylation of the amino-terminal tail of histone proteins are the most studied and best characterized ones. These modifications were the first histone PTMs discovered and were linked to altered rates of DNA transcription almost six decades ago[3,4]. Importantly, combinations of acetylation and methylation of lysine residues in histone tails can function in a concerted manner with both cooperative and antagonistic functions.

[1]Institut Pasteur, Université Paris Cité, CNRS UMR 6047, Biologie des Bactéries Intracellulaires, 75015 Paris, France. [2]Sorbonne Université, Collège doctoral, 75005 Paris, France. [3]Université Paris Cité, CNRS, Unité de Biologie Fonctionnelle et Adaptative, 75013 Paris, France. [4]Institut Pasteur, Université Paris Cité, CNRS UMR 3528, Unité de Microbiologie Structurale, 75015 Paris, France. [5]Present address: Herbert Wertheim School of Optometry & Vision Science, University of California, Berkeley, CA, USA. [6]Present address: Laboratory of Molecular & Structural Microbiology, Institut Pasteur de Montevideo, Montevideo, Uruguay. [7]Present address: Université Paris Cité, CNRS, UMR7126 Epigenetics and Cell Fate, 75013 Paris, France. ✉e-mail: cbuch@pasteur.fr; mrolando@pasteur.fr

Methylation, depending on the target residue, can be associated with compaction of chromatin and reduced transcription[5] whereas acetylation often impairs the affinity of histones to DNA, consequently loosening chromatin compaction, promoting the recruitment of transcription factors[6] and increasing the mobility of histones along the DNA[7]. Therefore, the balanced activity of enzyme classes involved in adding (histone methyltransferases and histone acetyltransferases (HAT)) and removing (histone demethylases and histone deacetylases (HDAC)) these groups ensure the correct expression of specific genes at specific times.

Numerous stimuli have been shown to influence the PTM levels of histones in the eukaryotic cell. Excitingly, it has been shown recently that different pathogens manipulate histone modifications mainly by secreting effector proteins targeting the nucleus, so-called nucleomodulins[8]. In particular, histone methylation and histone acetylation have been shown to be potent targets for a variety of pathogens to promote their replication in their host cells[9–12]. One of the pathogens described to modulate histone PTMs of its host cell is *Legionella pneumophila*, a facultative intracellular, Gram-negative bacterium that parasitizes free-living protozoa, but that is also the causative agent of a severe atypical pneumonia in humans, called Legionnaires' disease[13]. The intimate interaction of *L. pneumophila* with its eukaryotic hosts has shaped the bacterial genome and led to the evolution of numerous mechanisms allowing *L. pneumophila* to manipulate host functions and to thrive in this otherwise hostile intracellular environment. This *Legionella*-protozoa coevolution is particularly reflected in the presence of multiple genes encoding eukaryotic-like proteins in the *Legionella* genome[14–16]. Many of these proteins are secreted by the Dot/Icm type-IV secretion system (T4SS) that translocates more than 300 proteins into the host cell that are key for facilitating bacterial intracellular survival and replication[16–19]. One of these secreted eukaryotic-like effectors is a SET-domain histone methyltransferase, RomA, that mimics eukaryotic histone methyltransferases and directly modifies host chromatin by tri-methylating lysine 14 on histone H3 (H3K14)[20]. The unique activity of this bacterial protein leads to an increase in H3K14 methylation, thereby decreasing the expression of many genes involved in the host response to the infection[20]. Interestingly, H3K14me had not been identified in human cells before RomA activity was discovered but this modification was recently reported to be present at very low levels in human cells, probably explaining why it had been overlooked[21]. H3K14me increases under specific conditions such as stress response[22], however, the prevalent modification of H3K14 is acetylation.

Thus, the question emerges "How does the bacterial pathogen *L. pneumophila* methylate a histone residue that is usually acetylated?". Given the plethora of eukaryotic-like proteins in *L. pneumophila* we hypothesized that the bacteria might encode its own eukaryotic-like histone deacetylase (HDAC). Here we report that *L. pneumophila* encodes an HDAC-like protein (LphD) that works in synergy with RomA by specifically deacetylating H3K14 to facilitate methylation of H3K14 by RomA. Together these effectors function as effector pairs, or 'paraeffectors', with high levels of interdependence that serve to fine-tune the host cell's gene expression and promote bacterial intracellular replication.

## Results

### *L. pneumophila* encodes an HDAC-like protein that preferentially targets H3K14

Bioinformatic analyses of the *L. pneumophila* strain Paris genome identified the gene *lpp2163* encoding a 424 amino acid long protein, predicted to be a $Zn^{2+}$-dependent histone deacetylase, that we named LphD (*Legionella pneumophila* histone Deacetylase)[12]. We computed a structural model of LphD using AlphaFold[23] which allowed prediction of the protein structure with high confidence (Fig. 1A) except for the N- and C-terminal regions (amino acids 1-25 and 415-424), that are likely unstructured in solution. We sequence aligned a selection of eukaryotic HDACs (Fig. S1A) and used ConSurf[24] to compute the conservation scores and map them onto the LphD model (Fig. 1B). All the catalytic residues of the so-called charge relay system[25] including the active-site tyrosine (Y392) and the $Zn^{2+}$ coordinating atoms were conserved (Fig. 1B insert). An exception is N219 which is a histidine in eukaryotic structures but fulfills the same function in coordinating the zinc atom.

To determine if LphD indeed possesses lysine deacetylase activity, we performed a fluorometric enzymatic assay that allows the quantification of lysine deacetylation on an acetylated lysin side chain. Figure 1C shows that LphD exhibits in vitro lysine deacetylase activity in a dose-dependent manner. Furthermore, the single amino acid substitution (Y392F) at the predicted active site completely inactivates the enzyme. Adding Trichostatin A (TSA), a broad-range inhibitor of $Zn^{2+}$-dependent histone deacetylases, reduced LphD activity (Fig. 1C), further suggesting that LphD is a $Zn^{2+}$-dependent histone deacetylase. We then attempted to identify the targeted lysine(s) by using MS/MS analyses of histones extracted from THP-1 cells that had been incubated with purified LphD for 1 h at 37 °C. Several lysine residues were identified on H3, H4, and H2B (Figure S1B). However, as this approach did not use physiological quantities of LphD, it might not have revealed the specificity this protein could have during infection. Thus, we used AlphaFold to compute models of the complexes between LphD and the tails of histones H3 and H4 (residues 1-25) to see if we could get an indication for the physiological substrate specificity. In the case of H3, all 5 models show the same lysine (H3K14) binding to this pocket (Figure S1C, E). In contrast, for the H4 peptide the models were predicted with poor confidence, and in only 2 out of 5 complexes is a lysine (H4K9) residue positioned at the active site (Figure S1D, E). Furthermore, whereas most of the H3 peptide is predicted with poor confidence and away from the LphD surface, H3K14 is placed into the active site pocket, a tight cavity that accommodates H3K14 and shows room for an additional acetyl group (Fig. 1D, E). None of the other lysines of the peptide were predicted to bind LphD, probably due to specific substrate recognition through the flanking amino acids Pro12, Ala13, Gly15, and Gly16 (Fig. 1E inset). Based on the results of these models, H3K14 seems to be the preferred target of LphD.

We first analysed in vitro if LphD has indeed target specificity for H3K14 as predicted. Using histone H3 peptides acetylated on each lysine residues present on the H3 tail (H3K4, H3K9, H3K14, H3K18, H3K23, and H3K27) we measured the catalytic efficiency of LphD with varying peptide concentrations and with LphD-Y392F as control. As seen in Fig. 1F LphD has a high catalytic efficiency for H3K14 with an efficiency constant ($k_{cat}/K_M$) of 1.984 $μM^{-1}.min^{-1}$ as compared to a $k_{cat}/K_M$ of below 0.314 $μM^{-1}.min^{-1}$ for all other residues (Table 1). Deacetylase activity of LphD on H3K14 was also observed when testing histone nucleosomes and octamers isolated from human cells (Fig. 1G, S2A, S2B, S2C) with H3K14ac antibodies that we had validated by dot blot (Figure S2D). Indeed, the value of $k_{cat}/K_M$ is the highest for H3K14 and the deacetylation of H3K14 on octamers occurred already within 5 min, whereas for longer incubation times a loss of specificity was observed. This explains the MS/MS results obtained previously, where a one-hour incubation time led to a deacetylation of many residues and a loss of specificity.

Taken together, AlphaFold models, enzymatic assays on acetylated histone peptides, as well as the use of specific antibodies revealed that LphD has clear target specificity for H3K14.

### LphD is a secreted effector targeting host histones during infection

To characterize the function of LphD during infection we first assessed whether it is a T4SS substrate. THP-1 cells were infected with a *L. pneumophila* strain expressing a ß-lactamase-LphD fusion construct or the ß-lactamase-RomA (positive control), and translocation was

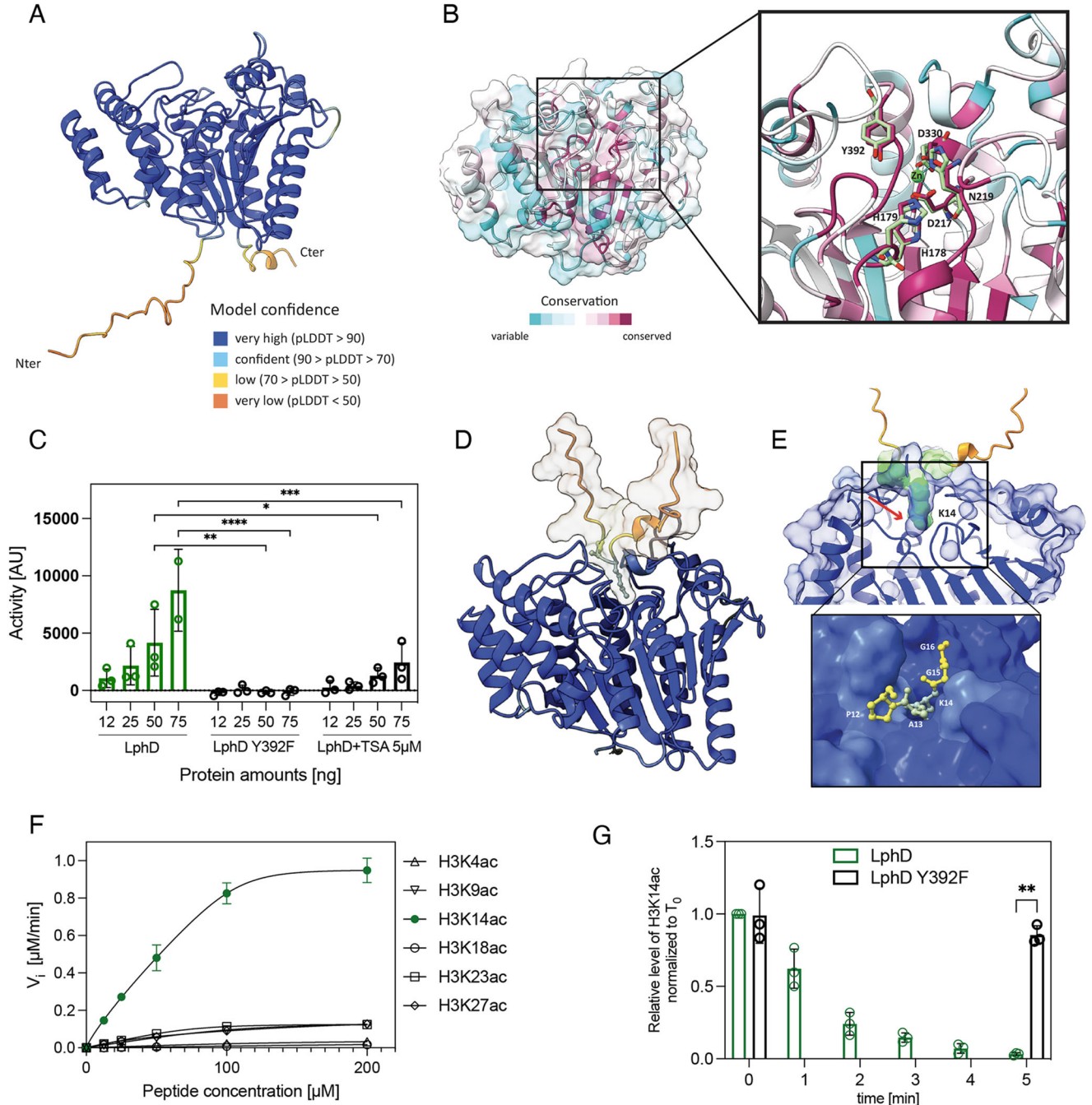

**Fig. 1 | LphD has predicted structural similarity to eukaryotic HDACs and possesses deacetylase activity. A** LphD model generated by AlphaFold v2.0.1. **Insert**: per-residue confidence score (pLDDT) produced by AlphaFold. **B** The conservation score between LphD and representative eukaryotic HDAC families mapped onto the LphD model. **Insert**: substrate binding groove with the catalytic residues (binding pocket residues and catalytic center) of LphD highlighted, colored according to conservation and HDAC6 colored green (PDB 5EDU). The catalytic zinc is shown from the HDAC6 structure. **C** Fluor de Lys® HDAC activity assay of LphD. Lysine deacetylase activity in vitro of increasing amounts of His-LphD and its catalytic inactive mutant (His-LphD Y392F). Control: 5 μM of Tri-chostatin A (TSA) (HDAC inhibitor). Data are presented as mean values ± SD of $n = 3$ independent experiments. For statistical analyses a two-way ANOVA was performed using Tukey's multiple comparisons test (*$p = 0.0486$, **$p = 0.027$, ***$p = 0.002$, ****$p < 0.001$). **D** LphD-H3 peptide complex generated by AlphaFold v2.0.1. The H3 peptide is represented as a surface/cartoon model, K14, A15, and P16

residues are shown as sticks. **E** H3K14 (in green) is positioned towards the active site and the active site cavity (in blue) shows space to accommodate an acetylated K14 residue (red arrow). **Insert**: Residues 12 to 16 of the H3 peptide are represented as ball and stick, the entrance of the active site of LphD as surface. **F** Steady-state kinetics of purified LphD on fluorescent H3 peptide substrates acetylated at different lysine residues (H3K4ac, H3K9ac, H3K14ac, H3K18ac, H3K23ac, H3K27ac). $V_i$ values (μM.min$^{-1}$) plotted against substrate peptide concentrations and curves fitted using Michaelis–Menten equation. Data are presented as mean values ± SD of $n = 3$ independent experiments. **G** Densiometry quantification of LphD and LphD Y392F activity on H3K14ac levels on nucleosomes. Time [minutes]: reaction stop. H3K14ac signal was quantified after immunoblot detection and normalized to 0 min. Data are presented as mean values ± SD of $n = 3$ independent experiments. Statistical analysis performed using paired t-test (**$p = 0.005139$). All source data of this Figure are provided as source file.

measured by quantifying the number of cells exhibiting ß-lactamase activity against a fluorescent substrate (CCF4). The secretion of ß-lactamase-LphD was clearly detected during infection. Moreover, LphD is secreted by the T4SS, as no ß-lactamase activity was observed when the ß-lactamase-LphD fusion protein was produced by a strain lacking a functional Dot/Icm secretion system (Δ*dotA*) (Fig. 2A, S3A). To assess the subcellular localization of LphD, we transiently

transfected HeLa cells with an EGFP-LphD fusion product, showing that it accumulates in the nucleus of transfected cells (Fig. 2B), compared to the cytosolic localization typically found for EGFP (Figure S3B). Furthermore, in LphD transfected cells a drastic decrease of the H3K14ac signal occurred (Fig. 2B), which was not seen in cells transfected with the Y392F mutant, although the LphD-Y392F catalytic inactive mutant was also located in the nucleus of transfected cells (Figure S3C). An antibody raised against LphD (validated in Figure S4A) confirmed that LphD accumulates in the host cell nucleus also during infection (Figure S4B). Furthermore, in Fig. 2C we observed an important decrease of the H3K14ac mark in cells where LphD targets the nucleus, compared to uninfected cells.

To assess the influence of LphD on the epigenetic status of H3K14 during infection, we isolated histones from cells infected with either *L. pneumophila* wild type or a Δ*lphD* strain and followed H3K14 acetylation as a function of time. Figure 3A shows that *L. pneumophila* wild type infection leads to a decrease in H3K14ac within 7 hours of infection, dependent on the presence of LphD, as the infection with the Δ*lphD* strain led to an increase in H3K14ac in the same timeframe (Fig. 3A, S4C). LphD activity during infection is specific for H3K14, as other tested residues (H3K18 and H3K23) did not show a difference whether the cells had been infected with the wild type or the Δ*lphD*

### Table 1 | Enzymatic constants of LphD activity against different acetylated peptides

| Substrate | $K_M$ [µM] | $k_{cat}$ [min⁻¹] | $k_{cat}/K_M$ [µM⁻¹·min⁻¹] |
|-----------|-----------|------------------|---------------------------|
| H3K4ac | 286.96 | 10.34 | 0.036 |
| H3K9ac | 143.15 | 28.06 | 0.196 |
| H3K14ac | 95.13 | 188.78 | 1.984 |
| H3K18ac | 3037.16 | 36.20 | 0.011 |
| H3K23ac | 73.74 | 23.22 | 0.314 |
| H3K27ac | 114.08 | 25.58 | 0.224 |

Kinetics constants ($K_M$ and $k_{cat}$) were individually calculated for each peptide from non-linear regression fit using OriginPro v8.0.

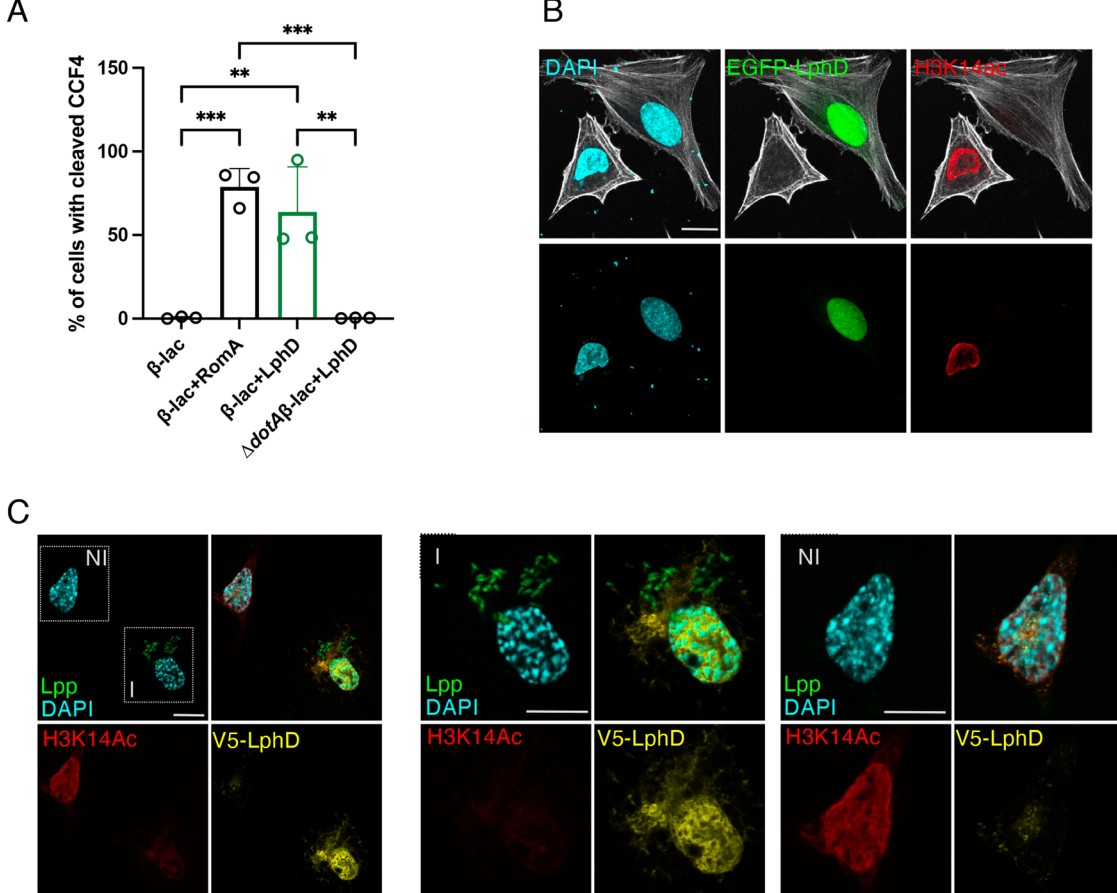

**Fig. 2 | LphD is a secreted effector that targets H3K14 in the nucleus. A** ß-lactamase secretion assays. Percentage of cells with cleaved CCF4 calculated as a ratio of secretion-positive (blue) over total stained cells (green and blue) after 2 hours of infection with *L. pneumophila* (*Lp*) wild-type (wt), or Δ*dotA* over-expressing either ß-lactamase (ß-lac), ß-lac fused to LphD (ß-lac+LphD) or to RomA (ß-lac+RomA). Data are presented as mean values ± SD of *n* = 3 independent experiments. For statistical analyses an ordinary one-way ANOVA was performed using Tukey's multiple comparisons test (**$p$ = 0.0032; ***$p$ = 0.0008). **B** Immunofluorescence analysis of EGFP-LphD and H3K14 acetylation. HeLa cells

were transfected with EGFP-LphD for 24 hours and stained for H3K14ac. DAPI (cyan), EGFP-LphD (green), H3K14ac (red), phalloidin (gray). Single-channel images are shown. Scale bars 10 µm. **C** Immunofluorescence analysis of LphD during infection. Differentiated THP-1 cells were infected 16 hours (MOI = 50) with *Lp* wt expressing V5-LphD and GFP. DAPI (cyan), V5-LphD (yellow), *Lp* (green), and H3K14ac (red). In the first panel uninfected (NI) and infected (I) cells are framed and zoomed in panels NI and I, respectively. Scale bars 10 µm. All source data of this Figure are provided as source file.

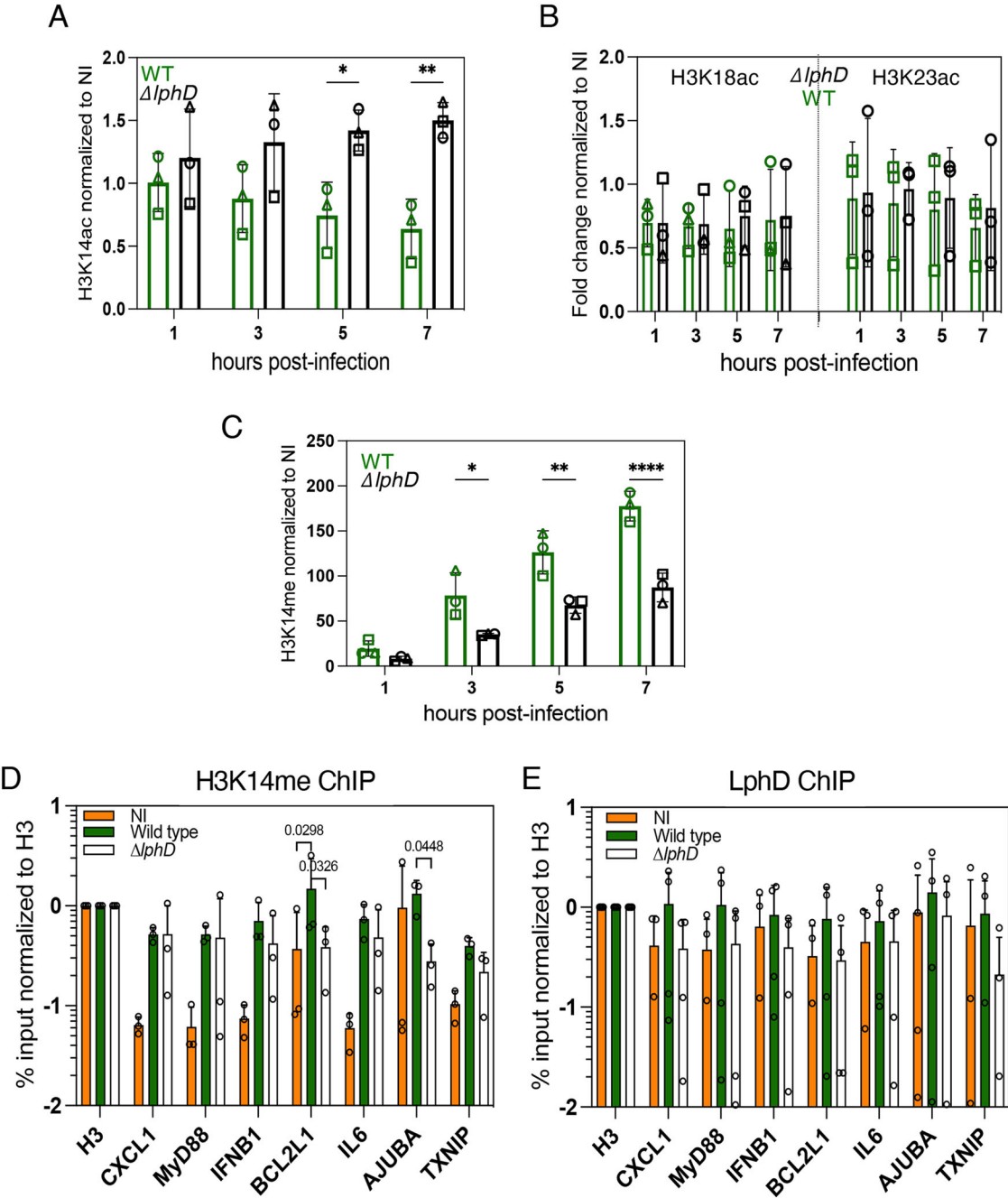

**Fig. 3 | LphD and RomA target H3K14 synergistically.** Western blot quantification of H3K14 (**A**), H3K18 or H3K23 acetylation (**B**). THP-1 cells infected with *L. pneumophila (Lp)* wild-type (wt) (green) or Δ*lphD* (white) expressing EGFP. Infected cells were enriched by FACS sorting (EGFP signal). Histones were isolated by acidic extraction and analysed by western blot. Loading control: Histone H1, signal is fold-change normalized to non-infected cells. Data are presented as mean values ± SD of *n* = 3 independent experiments. For **A** statistical analyses a two-way ANOVA was performed using Šidák's multiple comparisons test (**p* = 0.0317, ***p* = 0.0054). **C** Western blot quantification of H3K14 methylation. THP-1 cells were infected with *Lp* wt (green), Δ*lphD* (white) expressing EGFP. Infected cells were enriched by FACS sorting (EGFP signal). Histones were isolated by acidic extraction and analysed by western blot. Loading control: Histone H1, signal is fold-change normalized to non-infected cells. Data are presented as mean values ± SD of *n* = 3 independent

experiments. For statistical analyses a two-way ANOVA was performed using Šidák's multiple comparisons test (**p* = 0.0136, ***p* = 0.0010, *****p* < 0.0001). **D** ChIP of H3K14me during *Lp* infection followed by qPCR targeting selected promoters. THP-1 cells were uninfected (orange) or infected 7 hours with *Lp* wt (green) or Δ*lphD* (white) expressing EGFP. Infected cells were enriched by FACS (EGFP signal). Signal was normalized to histone H3. Data are presented as mean values ± SD of *n* = 3 independent experiments. For statistical analyses a two-way ANOVA was performed using Tukey's multiple comparisons test with **p* as indicated. **E** ChIP of LphD during *Lp* infection followed by qPCR targeting selected promoters. THP-1 cells were uninfected (orange) or infected with *Lp* wt (green) or Δ*lphD* (white) expressing EGFP. Infected cells were enriched by FACS (EGFP signal). Signal was normalized to histone H3. Data are presented as mean values ± SD of *n* = 4 independent experiments. All source data in this Figure are provided as source file.

mutant strain (Fig. 3B, S4D, E). Thus *L. pneumophila* encodes two effectors targeting the same lysine on the same histone tail, RomA that targets and methylates H3K14[20], and LphD that deacetylates H3K14. As we observed a decrease in H3K14 acetylation in presence of RomA, we

wondered whether this was due to a genome-wide H3K14me accumulation, or a specific and targeted deacetylase activity, possibly driven by the bacteria. Indeed, when LphD is absent (Δ*lphD*) the level of H3K14me during infection is significantly reduced from the early

stages of infection (1–3 hours) (Fig. 3C, S4F). To analyse whether LphD and RomA can act at the same promoters, we analysed promoters of genes we had previously shown to be the targets of H3K14 methylation by RomA by ChIP analyses[20]. This revealed that the methylation levels of H3K14 on specific promoters significantly decrease when infecting with a Δ*lphD* mutant strain where LphD deacetylation activity is missing, compared to wild-type infection (Fig. 3D). Methylation of H3K14 at these promoters, driven by RomA, is also helped by LphD binding to the same regions of the chromatin (Fig. 3E).

These results suggest that LphD directly impacts the activity of RomA on histones, and strongly indicate that the two effectors act in concert to modify the host chromatin landscape to downregulate host defense gene expression.

### LphD and RomA have complementary functions as virulence factors

To analyse whether LphD has an impact on intracellular replication of *L. pneumophila* we constructed a Δ*lphD* mutant strain by replacing the chromosomal gene of the wild-type strain by a gentamycin resistance cassette as previously described[20]. Secondly, *romA* was replaced in the Δ*lphD* strain to create a double mutant. All mutant strains were whole genome sequenced to ascertain the correct knock out of the gene and the absence of secondary mutations elsewhere. Replication of the wild-type strain compared to the Δ*lphD* mutant strain in macrophages derived from THP-1 cells as well as in *Acanthamoeba castellanii*, a natural host of *L. pneumophila*, was compared to determine whether LphD was important for intracellular growth of *L. pneumophila*. The Δ*lphD* strain showed a consistent growth delay compared to the wild-type strain in both THP-1 cells and in *A. castellanii* (Fig. 4A, B). Complementation of the Δ*lphD* strain with full-length *lphD* under the control of its native promoter completely reversed the growth delay and even induced a slight increase in replication due to the plasmid copy number, further underlining the role of LphD in virulence of *L. pneumophila* (Fig. 4C). When either LphD or RomA is deleted (Δ*lphD* and Δ*romA*) the bacteria show a defect in intracellular replication in THP-1 cells (Fig. 4D) and in *A. castallanii* (Fig. 4E). However, when the infection is performed with the double mutant Δ*lphD*Δ*romA* the phenotype is partially reversed (Figs. 4D, E), suggesting that the replication defect of the single knockout strains is not only caused by the absence of one effector, but also by the activity of the other effector alone. Hence, the advantageous influence of each effector on intracellular replication depends on the presence of the other. Moreover, this collaborative effect is dependent on the catalytic activity of each effector. Indeed, complementation of the double mutant with either *lphD* or *romA* alone shows the growth phenotype of the single mutant, which is not the case when complementing it with the catalytically inactive proteins (Fig. 4F). This dependence is also seen when the transcriptional response of THP-1 cells infected with *L. pneumophila* wild-type is compared to that of THP-1 cells infected with the Δ*lphD*, Δ*romA* and Δ*lphD*Δ*romA* knockout strains by RNAseq. In contrast to the Δ*romA* mutant, the transcriptional profile of the double knockout strain strongly resembles wild-type infected cells (Figure S5A).

Taken together, RomA activity strongly depends on the presence of active LphD on the transcriptional level as well as in intracellular replication. Given the strong interdependence of these two effectors we coined the term "para-effectors" for the RomA and LphD pair.

### LphD and RomA target host chromatin cooperatively

As LphD and RomA target the chromatin together, we first tested if the two effector proteins interact. In vitro binding assays with His- and GST-tagged proteins produced in *Escherichia coli* showed that RomA and LphD can indeed directly interact (Fig. 5A, S5B). Furthermore, we observed that when expressed in eukaryotic cells, RomA and LphD target the chromatin as they are enriched in the nuclear fraction (Fig. 5B, S5C). We thus searched for possible eukaryotic interacting partners or complexes that may facilitate chromatin targeting by these two bacterial effectors. To identify potential targets of LphD, we performed affinity chromatography by GFP-trap pull-down of HEK293T cells transfected with an EGFP-LphD construct followed by MS/MS analysis. We set a threshold for candidate proteins that at least two unique peptides were detected with a significant false discovery rate (<0.1) and a high (>4) fold change as compared to the control condition (GFP). This approach identified a total of 542 significantly enriched proteins, compared to the control (EGFP) (Fig. 5C). Interestingly, among this set of proteins, many of the identified peptides were derived from proteins that are known to be involved in epigenetic regulations of the cell (Table S1). Among those the histone acetyl transferase (HAT) KAT7 (HBO1) was a promising candidate; KAT7 is the enzymatic subunit of the so-called HBO1 complex, comprised of KAT7, BRPF1-3, ING4/5, and MEAF6. This complex is well-known to bind histone H3, regulating the acetylation of K14 (Fig. 5D)[26].

Using a GFP-trap to pull down EGFP-LphD and blotting for the binding of endogenous KAT7 we confirmed that KAT7 indeed interacts with endogenous EGFP-LphD, but not with EGFP alone (Fig. 5E). Importantly, the binding of LphD to KAT7 is independent of its enzymatic activity, as catalytically inactive LphD-Y392F still binds KAT7 (Fig. 5E, for IP control see Figure S5D). Reverse immunoprecipitation further verified that KAT7/LphD complex formation takes place in transfected cells (Figure S5E). Co-IP of EGFP-LphD and EGFP-RomA and further analyses of the presence of the different components of the HBO1 complex revealed that LphD interacts with all components of the HBO1 complex (BRPF1, KAT7, MEAF6 and ING5), whereas RomA binds only KAT7 in vitro (Fig. 6A, for IP control see Figure S5F). Importantly, we could concomitantly show that RomA and LphD both bind the HBO1 complex, as well as the target histone H3.

To further analyse these interactions during infection, we used HEK293T cells that stably express the macrophage Fcγ-RII receptor which allows to engulf bacteria that have been opsonized with antibodies, leading to an efficient infection in combination with high transfection efficiency[27]. We transfected these cells with either EGFP or EGFP-KAT7 and infected them using *L. pneumophila* wild-type strain overexpressing V5-tagged forms of LphD or RomA. When pulling down the EGFP proteins we clearly detected an enrichment of both LphD and RomA in the KAT7 expressing sample (Fig. 6B). In vitro binding assays with tagged forms of LphD, RomA and KAT7 produced in *E. coli* showed that RomA and LphD bind KAT7 also in a chromatin-free context (Fig. 6C). To exclude indirect interactions due to the presence of DNA we performed a GFP-trap to pull down EGFP-LphD and EGFP-RomA in the presence of ethidium bromide[28]. This confirmed, together with the in vitro results, that RomA and LphD interact with KAT7 (Figure S5G). Thus, binding of LphD and RomA to KAT7 is specific and independent of chromatin targeting. To further explore the role of KAT7 during *L. pneumophila* infection and the synergistic activities of LphD and RomA, we quantified the levels of H3K14 methylation during infection with *L. pneumophila* wild-type or the Δ*lphD* mutant strain by pre-treating the cells with a KAT7 specific inhibitor (WM-3835) (Fig. 6D). This revealed that inhibition of KAT7 activity abolishes the decrease in H3K14 methylation observed in infection with *L. pneumophila* lacking LphD (Δ*lphD*). Thus, LphD counteracts KAT7 activity (Fig. 6D and Figure S5H).

Taken together, these results further support our model that the two effectors, RomA and LphD, both target the KAT7 complex and chromatin to cooperatively regulate H3K14 acetylation and methylation for the benefit of intracellular growth of *L. pneumophila*.

## Discussion

Analyses of the *L. pneumophila* genome identified a secreted nucleo-modulin, named LphD, that possesses histone deacetylase activity. The study of the functional role of LphD in infection revealed the exceptional capacity of these bacteria to manipulate epigenetic marks of the

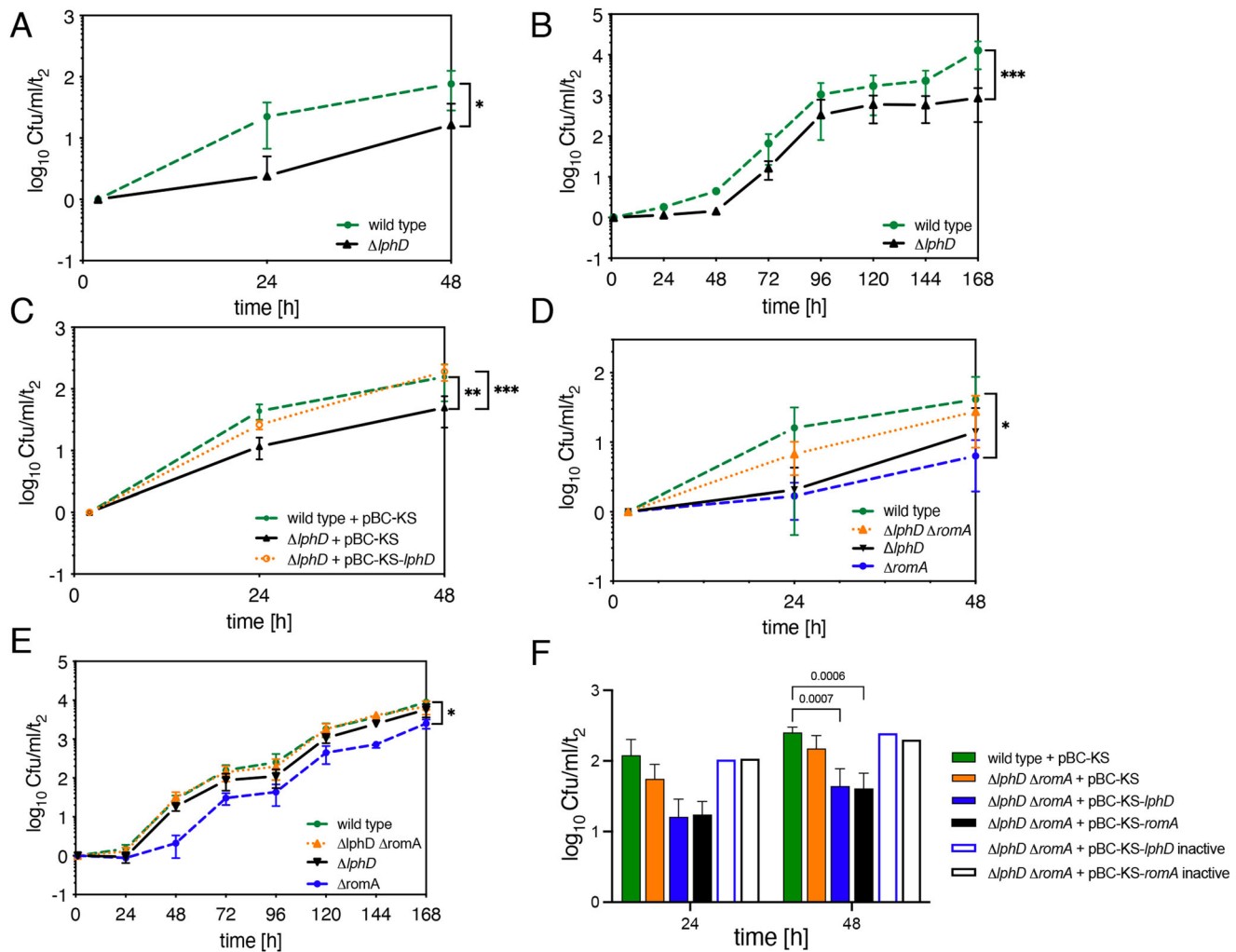

**Fig. 4 | LphD and RomA act as para-effectors during intracellular replication.** *L. pneumophila (Lp)* intracellular replication (log$_{10}$ ratio cfu/ml/t$_2$) in THP-1(MOI = 10 in **A**, **C**, **D**, **F)** or *A. castellanii* (MOI = 0.1 in **B**, **E**). **A** *Lp* wt (green), Δ*lphD* (black). Data are presented as mean values ± SD of *n* = 3 independent experiments. For statistical analyses a two-way ANOVA was performed using Šidák's multiple comparisons test (*\*p* = 0.0189). Non-linear regression analysis using a straight-line model showed a significant slope difference (*p* = 0.0291; *F* = 5.907). **B** *Lp* wt (green) and Δ*lphD* mutant (black). Data are presented as mean values ± SD of *n* = 4 independent experiments (except t24h and 48 h *n* = 2 independent experiments). For statistical analyses a two-way ANOVA was performed using Šidák's multiple comparisons test (*\*\*\*p* < 0.001). Non-linear regression analysis using a logistic growth model showed a significant difference in the curves (*p* < 0.001; *F* = 12.22). **C** Complementation analysis of *Lp* wt (green) and Δ*lphD (*black) with empty pBC-KS, Δ*lphD* with pBC-KS-*lphD* (orange). Data are presented as mean values ± SD of *n* = 4 independent experiments. For statistical analyses a two-way ANOVA was performed using

Tukey's multiple comparisons (*\*\*p* = 0.001; *\*\*\*p* < 0.001). **D** *Lp* wt (green), Δ*lphD*Δ*romA* (orange), Δ*lphD* (black), and Δ*romA* (blue). Data are presented as mean values ± SD of *n* = 4 independent experiments. For statistical analyses a two-way ANOVA was performed using Tukey's multiple comparisons test (*\*p* = 0.0189). **E** *Lp* wt (green), Δ*lphD* Δ*romA* (orange), Δ*lphD* (black), Δ*romA* (blue). Data are presented as mean values ± SD of *n* = 3 independent experiments. For statistical analyses a two-way ANOVA was performed using Tukey's multiple comparisons (*\*p* = 0.026). **F** Complementation analysis of *Lp* wt (green) and Δ*lphD*Δ*romA* (orange*)* with empty pBC-KS, pBC-KS-*lphD* (blue), and pBC-KS-*romA* (black); or pBC-KS-*lphD*-Y392F (blue, boxed) and pBC-KS-*romA*-Y249F/R207G/N210A (black boxed). Data are presented as mean values ± SD of *n* = 4 independent experiments (complemented *n* = 3, complemented inactive *n* = 2 independent experiments). For statistical analyses a two-way ANOVA was performed using Tukey's multiple comparisons test with *p* as indicated. All source data of this Figure are provided as source file.

host cell. Several bacterial effectors that target host chromatin have been identified in the last decade[8], however, the fact that different bacterial effectors may manipulate the host cell chromatin in synergy was unknown.

In eukaryotic cells, HDACs suppress gene expression by condensing chromatin packing and consequently decreasing chromatin accessibility for transcription factors and their binding to gene promoters[29]. The regulation of their activity is a complex, multilayer process depending, among others, on the subcellular localization and protein complex formation. Although several bacterial pathogens have been shown to impact histone acetylation/deacetylation during infection by acting on expression of eukaryotic HDACs or their localization[12], until now only the *Neisseria gonorrhoeae* Gc-HDAC

protein has been shown to induce an enrichment of H3K9ac, in particular at the promoters of pro-inflammatory genes[30].

Here we show that LphD encoded by *L. pneumophila* targets the host cell nucleus during infection, and deacetylates H3K14. The processes by which LphD targets the nucleus remains to be determined. Eukaryotic HDAC proteins are predominantly located in the nucleus, but some also shuttle between the nucleus and the cytoplasm, where they can also target cytoplasmic proteins. Class I HDACs whose subcellular localization is predominantly nuclear encode a nuclear localization sequence (NLS). Indeed, a putative NLS was predicted in silico at the N-terminus of LphD; however, its deletion did not affect nuclear translocation of LphD (Figure S6A). Thus, either another NLS is present in LphD that was not identified, or the nuclear import of LphD might be

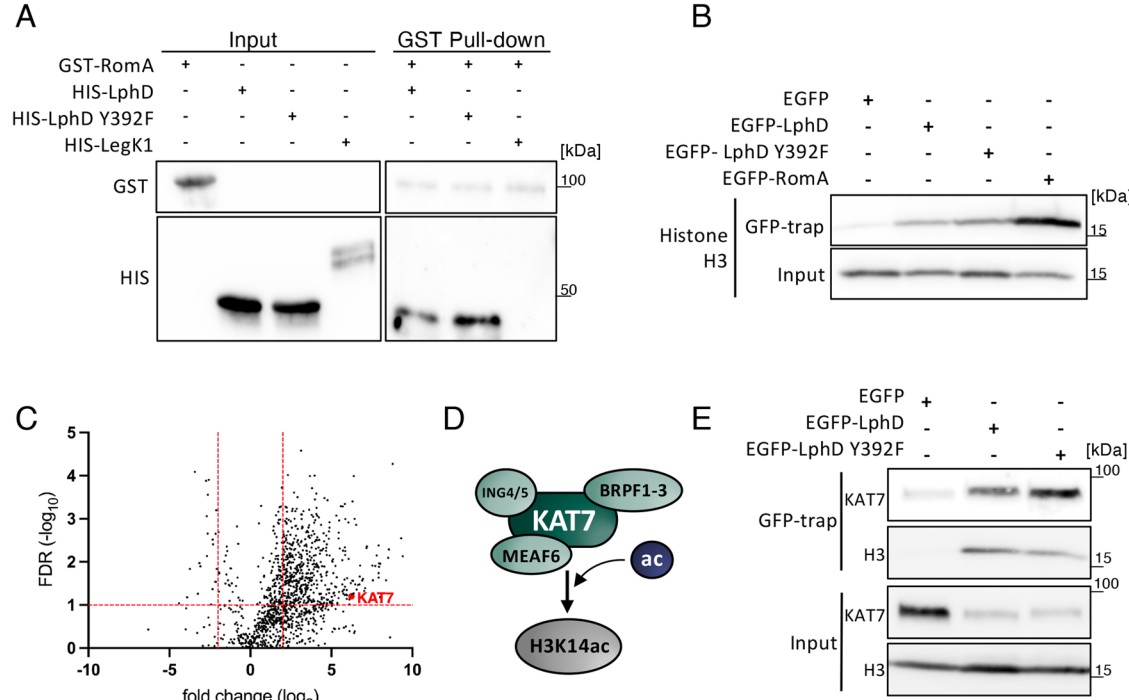

**Fig. 5 | LphD and RomA interact and target chromatin. A** In vitro protein interaction assay using purified GST-RomA and His-LphD. Purified proteins were mixed in equal amounts followed by a GST-pulldown using Glutathione-beads. HIS$_6$-LegK1 was used as negative binding control. Representative of $n = 3$ independent experiments. Source data provided as source file. **B** Immunoblots showing the interaction of LphD and RomA with histone H3. GFP-trap in HEK293T cells transfected with EGFP, EGFP-LphD, EGFP-LphD-Y392F, or EGFP-RomA. Pulldown samples were analyzed for the presence of H3. Input shows the expression level of endogenous histone H3 total lysate (IP control Figure S5B). Representative of $n = 3$ independent experiments. **C** Volcano plot of EGFP-LphD interacting proteins. The log$_2$ fold change of EGFP-LphD to control (GFP) is plotted against the −log$_{10}$ of the false discovery rate (FDR). Red: KAT7 selected for further analyses. Threshold: log$_2$ fold change >2 and FDR < 0.1 red lines. **D** Schema of the HBO1 histone acetyltransferase complex with BRPF1-3 targeting H3K14. **E** Immunoblots of LphD, KAT7, and histone H3 interaction. GFP-trap in HEK293T cells transfected with EGFP, EGFP-LphD, EGFP-LphD-Y392F. Input shows the expression level of endogenous KAT7 and Histone H3 in total lysates (IP control Figure S5D). Representative of $n = 3$ independent experiments. All source data of this Figure are provided as source file.

achieved by hijacking other HDAC-containing complexes such as NuRD. A key component of the NuRD complex, Retinoblastoma binding protein 4 (RBBP4), has been shown to directly interact with importin-α and to regulate the nuclear import[31,32]. We identified several components of the NuRD complex as possible interaction partners of LphD, suggesting that LphD hijacks the NuRD complex to translocate to the host cell nucleus (Table S1), however, further experiments are necessary to confirm this hypothesis.

In vitro activities on short H3 peptides acetylated on different lysine residues showed a clear preference of LphD for H3K14 (Fig. 1F), supported by an AlphaFold model of the complex between LphD and the histone H3 peptide tail, where H3K14 is placed in the active site pocket (Fig. 1E and inset). Thus, H3K14 is the residue for which LphD exhibits the highest enzymatic efficiency, but the enzyme loses specificity if the reaction is allowed to proceed for a long time or when high concentrations of enzyme are present. This is seen in transfection when high amounts of protein are delivered into a host cell, as LphD transfection in HeLa cells leads to the deacetylation of several other residues such as H3K18 and H3K23 (Figure S6B). In contrast, during infection, when the enzyme is delivered in the host cell at physiological levels, only H3K14 is deacetylated (Fig. 3A and B).

The deacetylation of H3K14 facilitates the methylation of the same residue by another bacterial effector, the SET-domain methyltransferase RomA. Importantly, we observed here that LphD and RomA not only target the same host protein, but that RomA activity is dependent on the presence of LphD as the level of H3K14 methylation during infection is significantly reduced in a *ΔlphD* background (Fig. 3C). The cooperation between LphD and RomA was also observed in intracellular infection models and reflected in the transcriptome

profiles (Fig. 4 and S5A). The dependency of the two effectors on each other is even more striking in the phenotype of the double knockout. In replication assays, the *ΔlphDΔromA* strain shows a phenotype closely resembling the wild-type, compensating for the effect observed in the two single knockouts. Moreover, this compensation is reversible by complementation of either LphD or RomA, an effect not seen when using the catalytically inactive versions of the proteins for complementation analyses. This is an intriguing finding and prompted us to search the literature for similar phenotypes. Indeed, Liu and colleagues showed that in *Escherichia coli*, the knockout of the genes *relA* or *dnaK* causes a decrease in persister formation compared to wild-type bacteria in the presence of ampicillin. However, a double knockout strain (*ΔrelA ΔdnaK*) showed a higher persister formation than the two single knockouts, closely resembling the wild-type phenotype[33]. The exact mechanism of this phenotype is unknown.

We do not know either the exact mechanism but hypothesise that the phenotype of the *ΔlphDΔromA* double knock out strain could be due to specific regulations. In the double knock out the bacterial cell might compensate for the impact on replication by expressing other proteins involved in the intracellular cycle of *L. pneumophila* thereby counteracting the host response in a different way to allow effective replication. Indeed, it is known that the *Legionella* genomes encode for redundant effector functions. For modulating one host pathway a number of different effectors acting on same and/or different steps of the same pathway may be present[34]. Another possibility is that the absence of one of the two bacterial effectors, and the subsequent suboptimal modification of H3K14 triggers the host cell response, leading to an impaired bacterial replication. If both effectors are missing, this "incomplete" modification does not occur, hence there is

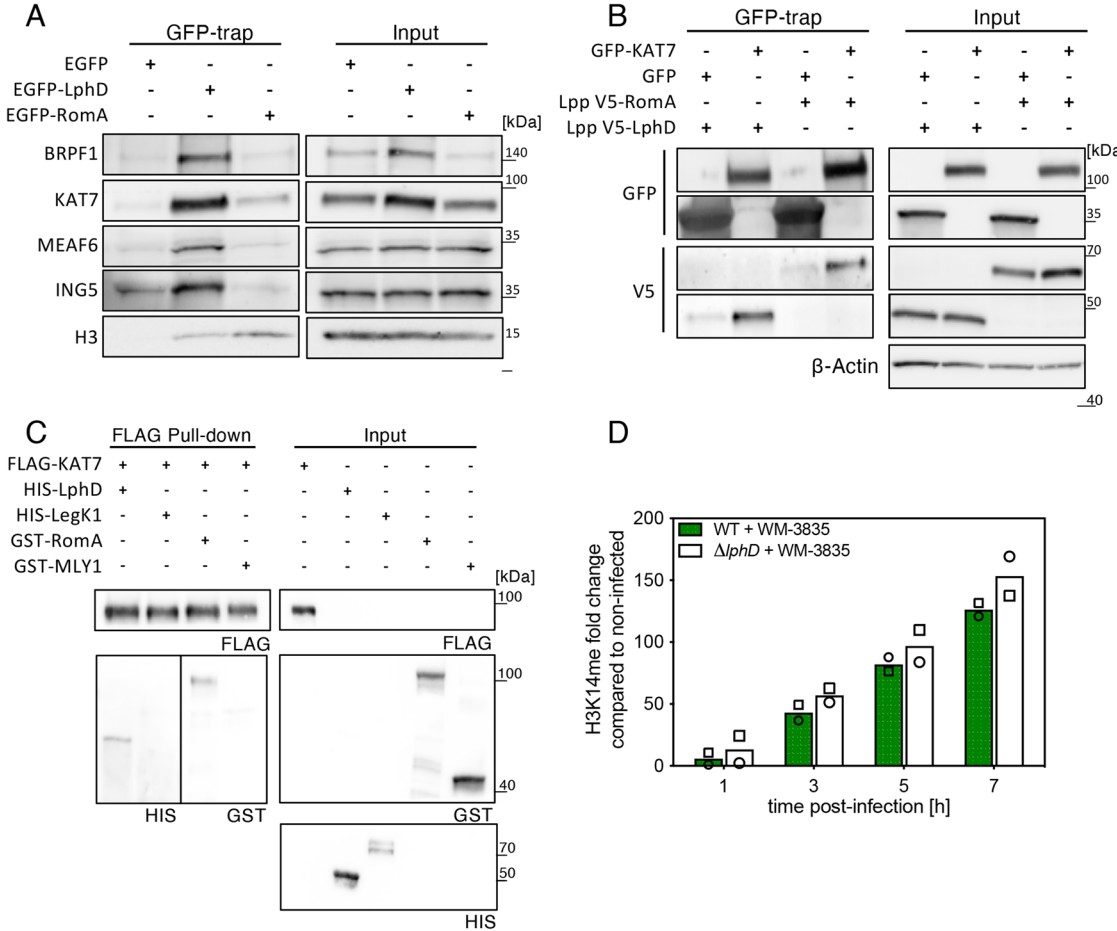

**Fig. 6 | LphD and RomA target chromatin via the HBO1/KAT7 complex.**
**A** Immunoblots showing the interaction of LphD and RomA with the HBO1 complex and histone H3. GFP-trap in HEK293T cells transfected with EGFP, EGFP-LphD, or EGFP-RomA. Samples were analyzed for the presence of the different components of the HBO1 complex (BRPF1, KAT7, MEAF6, ING5). Input shows the expression level of endogenous HBO1 components and histone H3 in total lysates. For IP control see Figure S5F. Representative of $n = 3$ independent experiments. **B** Immunoblots showing the interaction of LphD and RomA with KAT7. GFP-trap in HEK293-FcγRII cells transfected with either EGFP or EGFP-KAT7 followed by infection with *L. pneumophila* (*Lp*) wild-type (wt) over-expressing either V5-LphD or V5-RomA. EGFP proteins were pulled down using GFP-trap beads and samples were analyzed for the presence of V5-LphD or V5-RomA. Input shows the expression level in total lysates

of transfected EGFP-KAT7, V5-fusion proteins, as well as ß-actin (loading control). Representative of $n = 3$ independent experiments. **C** In vitro protein interaction assay using purified FLAG-KAT7, His-LphD, and GST-RomA. Purified proteins were mixed in equal amounts followed by a FLAG-pulldown using FLAG-trap beads. His-MBP and GST-MYL1 were used as negative binding controls. Representative of $n = 3$ independent experiments. **D** Quantification of western blot signal for H3K14 methylation in WM-3835 (KAT7-specific inhibitor) treated cells. THP-1 cells were treated with WM-3835 (1 μM) 18 hours prior to infection with *Lp* wt (green) or Δ*lphD* (white) expressing EGFP. Cells were sorted at different times post-infection by FACS (EGFP signal), histones were isolated and analysed by western blot. Loading control: Histone H1, compared to non-infected cells ($n = 2$ independent experiments). All source of this Figure are data provided as source file.

no trigger for the host cell. However, further experiments and analyses are necessary to completely understand the mechanism leading to this phenotype.

This suggests that the enzymatic activities of LphD and RomA are only beneficial for the bacteria when both partners are present through a mechanism of regulation that is yet to be understood. Thus, RomA and LphD are "para-effectors", from the Greek παρα (para) meaning *besides* but also *contrary to*, underlining the high interdependence of these two effectors. At least three *L. pneumophila* effectors have been described to target the same host protein, the small GTPase Rab1 and to sequentially modify it[35]. However, uniquely we show here that the lack of the activity of one effector decreases the activity of the other one and both target the endogenous chromatin binding complex KAT7/HBO1 (Figs. 5 and 6). Indeed, LphD and RomA immunoprecipitated host chromatin with an affinity for KAT7, as confirmed by in vitro binding assays. However, the fact that inhibition of KAT7 reverts the phenotype seen on H3K14me levels during Δ*lphD* infection might also imply that LphD directly modifies KAT7 acetylation. To explore the hypothesis that LphD also deacetylates KAT7 to

modulate its activity, further experiments will be undertaken. Indeed, how KAT7 activity is regulated in a cell is still not known. Interestingly, KAT8, a close homologue of KAT7 regulates its activity through autoacetylation of a lysin residue that is also conserved in KAT7, also leaving the possibility that KAT7 regulation functions in the same manner[36] and suggesting that KAT7 might be a non-histone target of LphD.

LphD might have complex roles in the host cell and be implicated in regulating several histone and non-histone targets as the analysis of possible eukaryotic interaction partners of LphD led not only to the identification of KAT7/HBO1, but also suggested interaction with several other complexes related to chromatin remodeling. Some of these complexes, such as the NuRD- and the Sin3-complex, are known to comprise eukaryotic histone deacetylases and might thus be additional interaction partners for LphD[37,38]. Further studies will elucidate whether additional interaction partners of LphD and RomA exist, that could be important during infection of different host cells, as *L. pneumophila* is known to infect many protozoan hosts and different mammalian cells. In conclusion, this study provides exciting insight

into how *L. pneumophila* modifies host chromatin by using two distinct chromatin remodelers that function in synergy. Both LphD and RomA are deployed by the bacteria to strategically influence the response of the host cell to infection and promote bacterial replication in this otherwise hostile environment.

# Methods

## Bacterial strains, growth conditions, and cell culture
*Legionella pneumophila* strain Paris and mutants were cultured in N-(2-acetamido)−2-aminoethanesulfonic acid (ACES)-buffered yeast extract broth (BYE) or on ACES-buffered charcoal-yeast (BCYE) extract agar[39]. For *Escherichia coli* Luria-Bertani broth (LB) was used. When needed antibiotics were added: for *L. pneumophila* (*E. coli*): kanamycin 12.5 μg/ml (50 μg/ml), gentamycin 12.5 μg/ml, chloramphenicol 10 μg/ml (20 μg/ml), and ampicillin (only for *E. coli*) 100 μg/ml.

THP-1 cells (ATTC™: TIB-202) were maintained in RPMI-1640+GlutaMAX (Gibco), HeLa (ATTC™: CCL-2), HEK293T (ATTC™: CRL-3216) and HEK-293 cells stably expressing the FcγRII receptor (gift of Prof. Craig Roy[27]) in DMEM + GlutaMAX (Gibco), both containing 10% FBS (Eurobio Scientific) in a humid environment with 5% CO$_2$ at 37 °C. HEK293T and the HEK-293 cells stably expressing the FcγRII receptor were not authenticated. However, the only entry of misidentified HEK cells at the ICLAC database is ICLAC-00063, when HEK cells were misidentified in 1981 with HeLa cells. We identified the HEK cells used by morphology and attachment as HEK cells are morphologically different from HeLa cells and they attach very weakly to tissue culture-treated plastic, in contrast to HeLa cells. Most importantly, the remote possibility of misidentification with HeLa cells would not change any consideration or conclusion in our study. Before use, the cells were tested negative for *Mycoplasma* contamination. *Acanthamoeba castellanii* [ATCC50739] was cultured at 20 °C in PYG 712 medium [2% proteose peptone, 0.1% yeast extract, 0.1 M glucose, 4 mM MgSO$_4$, 0.4 M CaCl$_2$, 0.1% sodium citrate dihydrate, 0.05 mM Fe(NH$_4$)$_2$(SO$_4$)$_2$•6H$_2$O, 2.5 mM NaH$_2$PO$_3$, 2.5 mM K$_2$HPO$_3$]. Cell transfections were performed by using FuGENE (Promega) following the recommendations of the manufacturer.

## Constructions of mutants and complementation plasmids
The knockout of *lphD* in the wild-type background to generate a single mutant followed by the knockout of *romA* to generate the double mutant, was performed as previously described[40]. To construct the Δ*lphD* mutant strain the chromosomal gene *lphD* of the wild-type strain was replaced by introducing a gentamycin resistance cassette. The mutant allele was constructed using a 3-steps PCR. Briefly, three overlapping fragments (*lphD* upstream region- primers 195H and 196H, antibiotic cassette-primers 52H and 52B, *lphD* downstream region- primers 195B and 196B; Table S2) were amplified independently and purified on agarose gels. The three resulting PCR products were mixed at the same molar concentration (15 nM) and a second PCR with flanking primer pairs (primers 195H and 195B; Table S2) was performed. The resulting PCR product, the gentamycin resistance cassette flanked by 500 bp regions homologous to *lphD* was introduced into strain *L. pneumophila* Paris by natural competence for chromosomal recombination. Strains that had undergone allelic exchange were selected by plating on BCYE containing gentamycin and the mutant was verified by PCR and sequencing. The resulting mutant was then used to generate the Δ*lphD* Δ*romA* double knockout strain, using a new set of primers (*romA* upstream region- primers 11H and 66H, kanamycin antibiotic resistance cassette-primers 60H and 60B, *romA* downstream region- primers 11B and 66B; Table S2). The generated mutant strains were whole genome sequenced to confirm the correct deletion of the gene and the absence of other mutations in the genome. For construction of the the complementation vector, the full-length *lphD* with its own promotor was cloned into pBC-KS (Stratagene). Bacteria expressing EGFP were obtained by introducing EGFP

under the control of the *flaA* promotor of *L. pneumophila* into pBC-KS backbone. To generate the catalytic inactive Y392F mutant of LphD, a single base pair mutation was performed using mismatched primers (217H and 217B, Table S2).

## ß-lactamase translocation assays
ß-lactamase assays were performed in THP1 infected cells as previously described[41]. Around 10$^5$ THP-1 cells were seeded in a 96-well plate and differentiated for 72 hours using 50 μg/ml phorbol 12-myristate 13-acetate (PMA). One day before infection bacterial strains (*L. pneumophila* wild-type or a Δ*dotA* mutant) carrying plasmids for the expression of either ß -lactamase alone or a ß -lactamase-LphD or -RomA fusion proteins were cultured in BYE broth containing chloramphenicol and IPTG (1 mM) to induce protein production. Cells were infected with the ß -lactamase fusion protein-expressing bacteria at an MOI of 50 in presence of 1 mM IPTG. Two hours after infection, Live-BLAzer CCF4-AM solution (Thermo Fisher Scientific) was added, and the plate incubated in the dark at room temperature for 2 hours. The cells were washed and detached by addition of cell dissociation solution (SIGMA) prior to flow cytometry analyses (MACS Quant, Miltenyi Biotec). FlowJo™ v10.8 Software was used for plot analyses (BD Life Sciences). For gating strategy see Figure S3A.

## Infection Assays
*A. castellanii* were washed once with infection buffer (PYG 712 medium without proteose peptone, glucose, and yeast extract) and seeded at a concentration of 4 × 10$^6$ cells per T25 flask. *L. pneumophila* wild-type and mutant strains were grown on BCYE agar to stationary phase, diluted in infection buffer and mixed with *A. castellanii* at an MOI of 0.1 or 1 (for complementation assays). Infected cells were maintained at 20 °C and intracellular multiplication was monitored plating a sample at different time points (2 h, 24 h, 48 h, 72 h, 96 h, 120 h, 144 h, and 168 h) on BCYE plates and the number of intracellular bacteria was counted. In THP-1 cell infection assays 10$^6$ cells were split in conical tubes (Falcon, BD lab ware). Stationary phase *L. pneumophila* were resuspended in serum free medium and added to cells at an MOI of 10. After 1 hour of incubation, Gentamycin (100 μg/ml) was added. After another hour of incubation, the infected cells were washed with PBS, before incubation with 2 ml of RPMI. At 2 h, 24 h, and 48 h, 500 μl of cell suspension were mixed with equal amounts of PBS-0.2% TritonX-100 for lysis. The infection efficiency was monitored by determining the number of colony-forming units (cfu) of the different *L. pneumophila* strains after plating on BCYE agar.

## Immunofluorescence analyses
For immunofluorescence analyses, cells are fixed with PBS-4% paraformaldehyde for 15 minutes at room temperature, followed by quenching (PBS-50 mM NH$_4$Cl) for 10 minutes. Cells are permeabilized with PBS-0.1% Triton X-100 and blocked for 30 minutes with PBS-5% BSA. The cells are incubated with the respective primary antibodies overnight at 4 °C in PBS-5% BSA. They are washed three times using PBS and then stained with DAPI, phalloidin, and secondary antibodies for 30 minutes at room temperature, followed by mounting to glass slides using Mowiol (SIGMA). Antibodies used in this study are listed in Table S3. Immunosignals were analyzed with a Leica SP8 Microscope at 63× magnification. Images were processed using Fiji software[42].

## LphD purification
N-terminal HIS$_6$-tagged LphD was expressed in *E. coli* BL21 C41 following an auto-induction protocol[43]. After 4 hours at 37 °C cells were grown for 20 hours at 20 °C in 2YT complemented autoinduction medium containing 50 μg/ml kanamycin. Cells were harvested and flash-frozen in liquid nitrogen. Cell pellets were resuspended in 50 ml lysis buffer (50 mM HEPES pH8, 500 mM KCl, 5% glycerol, 1 mM MgCl$_2$, benzonase, lysozyme, 1 mM DTT and supplemented with EDTA-free

protease inhibitor cocktails (ROCHE) at 4 °C and disrupted by sonication (6 × 60 s). The lysate was centrifuged for 60 min at 10,000 $g$ at 4 °C. The cleared lysate was loaded onto a Ni-NTA affinity chromatography column (HisTrap FF crude, GE Healthcare) equilibrated in Buffer A (50 mM Hepes pH8, 500 mM KCl, 5% glycerol, 10 mM imidazole, 1 mM DTT). HIS$_6$-tagged proteins were eluted with a linear gradient of buffer B (50 mM Hepes pH8, 500 mM KCl, 5% glycerol, 1 M imidazole, 1 mM DTT). The eluted fractions containing the protein of interest were pooled and dialysed at 4 °C overnight in SEC buffer (20 mM HEPES pH8, 300 mM KCl, 5% glycerol, 2 mM TCEP). The HIS$_6$-tag was not removed as this led to precipitation of the protein. After dialysis, the protein was concentrated and loaded onto a Superdex 75 16/60 size exclusion (SEC) column (GE Healthcare). The peak corresponding to the protein was concentrated to about 12 mg/ml and flash frozen in liquid nitrogen and stored at −80 °C.

### UFLC-mediated LphD deacetylase activity assay
In order to quantify LphD deacetylase activity, we synthetized six 5-fluorescein amidite (5-FAM)-conjugated acetylated peptide substrates based on the human H3.1 sequence and centered on various lysine residues of interest: ARTK$_{ac}$QTARRSK-(5-FAM), referred to as H3K4ac peptide; (5-FAM)-QTARK$_{ac}$STGG-NH$_2$, referred to as H3K9ac peptide; (5-FAM)-STGGK$_{ac}$APRR-NH$_2$, referred to as H3K14ac peptide; (5-FAM)-RAPRK$_{ac}$QLAT-NH$_2$, referred to as H3K18ac peptide; (5-FAM)-QLATK$_{ac}$AARR-NH$_2$, referred to as H3K23ac peptide; (5-FAM)-TRAARK$_{ac}$SAPAT-NH$_2$, referred to as H3K27ac peptide. Non-acetylated versions of these peptides were also synthetized as detection standards. Samples containing H3 peptides, and their acetylated forms were separated by RP-UFLC (Shimadzu) using Shim-pack XR-ODS column 2.0 ×100 mm 12 nm pores at 40 °C. The mobile phase used for the separation consisted of the mix of 2 solvents: A was water with 0.12% trifluoroacetic acid (TFA) and B was acetonitrile with 0.12% TFA. Separation was performed by an isocratic flow depending on the peptide: 83% A/17% B, rate of 1 ml/min, 6 min run for H3K4ac peptide; 80% A/20% B, rate of 1 ml/min, 6 min run for H3K9ac, H3K14ac, H3K18ac, H3K27ac peptides; 79% A/21% B, rate of 1 ml/min, 8 min run for H3K23ac peptide. H3 acetylated peptides (substrates) and their non-acetylated forms (products) were monitored by fluorescence emission ($\lambda = 530$ nm) after excitation at $\lambda = 485$ nm and quantified by integration of the peak absorbance area, employing a calibration curve established with various known concentrations of peptides. The kinetic parameters of LphD on H3-derived peptides were determined by UFL in a 96-wells ELISA. Briefly, LphD (7.7 nM) was mixed with different concentrations of acetylated H3 peptides (ranging from 12.5 to 200 μM final) for 15 minutes at 30 °C and the reaction was stopped by adding 50 μL of HClO$_4$ (15% v/v in water). Finally, 10 μl of the reaction mix were automatically injected into the RP-UFLC column and initial velocities (V$_i$, μM.min$^{-1}$) were determined as described above. V$_i$ were then plotted against substrate peptide concentrations and curves were non-linearly fitted using Michaelis–Menten equation $\frac{V_m*[S]}{K_m+[S]}$ (OriginPro v8.0). K$_m$ (enzyme Michaelis's constant), V$_m$ (enzyme maximal initial velocity), and k$_{cat}$ (enzyme turnover) values were extrapolated from these fits. A catalytic dead version of the enzyme was used as a negative deacetylation control[44].

### Highly acetylated histone extraction
HEK293T cells were cultivated in RPMI 1640 medium with 10% heat-inactivated fetal bovine serum (FBS) and 1 mM L-glutamine at 37 °C under 5% CO$_2$. For endogenous histone extraction, cells were seeded at 30,000 cells/cm² in a 100 cm² Petri dish (VWR). The next day, cells were treated with 20 mM sodium butyrate and 6 μM Trichostatin A (TSA). Cells were then put back in the incubator at 37 °C and 5% CO$_2$ for 30 min before being harvested. Cells were lysed with cell lysis buffer (PBS 1x, 1% Triton X-100, 20 mM sodium butyrate, 6 μM TSA, protease inhibitors) for 30 min at 4 °C, sonicated (2 s, 10% power) and

centrifuged (15 min, 15,500 g, 4 °C). 500 μL of 0.2 N HCl was then put on remaining pellets. The mixture was sonicated 3 times (3 s, 10% power) and incubated overnight at 4 °C. The next day, samples were centrifuged (15 min, 15,500 g, 4 °C) and the supernatant (containing extracted histones) was buffer exchanged three times into Tris 50 mM, 50 mM NaCl, pH 8 using MiniTrap G-25 desalting columns (GE Healthcare) and stored with protease inhibitor at −20 °C until use.

### In vitro histone deacetylase activity assay
Assays were performed in a total volume of 20 μL LphD buffer containing 250 ng highly acetylated, purified histones and 10 ng of LphD (wild-type or catalytic inactive mutant LphD- Y392F) at 30 °C. At different time points (0, 1, 2, 3, 4, and 5 minutes), the reaction was stopped with the addition of 10 μL Laemmli sample buffer containing 400 mM β- mercaptoethanol. Samples were analyzed on 18% SDS PAGE, followed by a transfer onto a nitrocellulose membrane (0.2 μm) at 200 mA for 65 min. Ponceau staining was carried out to ensure equal protein loading. Membranes were blocked with non-fat milk (5%) in PBS-1% Tween (PBST) for 1 h and incubated with α-H3K14Ac antibody in 1% non-fat milk PBST overnight at 4 °C. After washing 3 times, the membranes were incubated for 1 h at room temperature with peroxidase-coupled secondary antibody in 1% non-fat milk PBST. The proteins were then visualized by chemiluminescence detection using ECL reagent on LAS 4000 instrument (Fujifilm). Images were processed and quantified using MultiGauge v3.0 and Fiji software.

### In vitro nucleosome deacetylase activity assay
Assays were performed in a total volume of 20 μL Tris 20 mM, 150 mM NaCl, pH 7.4, 2 mM DTT buffer containing 250 ng K14-acetylated recombinant mono-nucleosomes (#81001, Active motif) and 10 ng LphD (WT or Y392F mutant) at 30 °C. At different time points (0, 1, 2, 3, 4, and 5 minutes), the reaction was stopped with the addition of 10 μL Laemmli sample buffer containing 400 mM β-mercaptoethanol. Samples were analyzed on gradient 4–12% SDS PAGE, followed by a transfer onto a nitrocellulose membrane (0.2 μm) at 210 mA for 65 min. Ponceau staining was carried out to ensure equal protein loading. Membranes were blocked with non-fat milk (5%) in PBS with 1% Tween (PBST) for 1 h and incubated either with α-H3K14Ac antibody (1:10000, #ab52946 Abcam) in 1% non-fat milk PBST over night at 4 °C. After washing 3 times, the membranes were incubated for 1 h at room temperature with peroxidase-coupled secondary antibody in 1% non-fat milk PBST. The proteins were then visualized by chemiluminescence detection using ECL reagent on LAS 4000 (Fujifilm) instrument. Membranes were finally stripped and reprobed with α-H3 antibody (1:10000, #3638 Cell Signaling) following the same protocol than before. Images were processed and quantified using ImageJ software (v2.9.0/1.53t).

### Fluor de Lys® in vitro enzymatic assays
Purified HIS$_6$-LphD was used to perform in vitro enzymatic assays with Fluor de Lys® deacetylase assay (Enzo Life Sciences). Briefly, different amount of purified LphD and catalytic inactive mutant LphD Y392F were incubated with the substrate in the reaction buffer (20 ml; 50 mM TRIS/Cl, pH 8.0, 137 mM NaCl, 2.7 mM KCl, 1 mM MgCl$_2$) for 30 minutes at 37 °C. The signal was then read using a plate reader (TECAN). Trichostatin-A (TSA) was added at 5 μM.

### Histone modification analysis of infected cells
For the analysis of histone modifications during infection, THP-1 cells were infected at an MOI of 50 with *L. pneumophila* wild-type or mutant strains, both containing a plasmid for the expression of EGFP under the control of the *flaA* promoter. After 30 minutes, Gentamicin was added (100 μg/ml) to kill extracellular bacteria. Cells are then sorted by FACS (S3e, BIORAD) as previously described[45]. Histones of infected cells were isolated as previously described with some modifications[46].

 

Briefly, THP-1 cells ($3 \times 10^6$) were incubated at 4 °C with hypotonic lysis buffer (10 mM Tris–HCl pH 8.0, 1 mM KCl, 1.5 mM MgCl$_2$, with protease inhibitor cocktail [ROCHE]) for 2 hours while rotating. Subsequently, nuclei were pelleted and resuspended in 0.4 M sulfuric acid and incubated overnight at 4 °C. The supernatant was precipitated with 33% trichloroacetic acid (TCA) on ice. Pelleted histones were washed twice with ice-cold acetone and were then resuspended in DNase/RNase free water. Sample quality of acid extraction was visualized on a Coomassie-stained 4-15% SDS-PAGE. Histone modification signal (H3K14ac, H3K14me, H3K18ac, H3K23ac) is assessed by immunoblot and normalized to signal of histone H1. Sample signals are then compared to non-infected controls.

## Small-scale biochemical fractionation
For small-scale biochemical fractionation[47] $4 \times 10^5$ HEK293T cells were transiently transfected with 1 μg of either EGFP, EGFP-LphD (24 hours), or LphD-Y392F, EGFP-RomA (48 hours), then washed with phosphate-buffered saline (PBS) and resuspended in buffer A (10 mM HEPES [pH 7.9], 10 mM KCl, 1.5 mM MgCl2,0.34 M sucrose, 10% glycerol, 1 mM dithiothreitol, 0.1% Triton X-100 and protease inhibitor cocktail [ROCHE]) (T: total fraction). The cells were incubated on ice for 10 min, and nuclei collected by centrifugation (5 min, 1300 g, 4 °C). The P1 nuclei were washed once in buffer A and lysed for 30 min in buffer B (3 mM EDTA, 0.2 mM EGTA, 1 mM dithiothreitol, and protease inhibitor cocktail [ROCHE]), and insoluble chromatin (fraction P3) and soluble (fraction S3) fractions were separated by centrifugation (5 min, 1300 g, 4 °C). The P3 fraction was washed once with buffer B and resuspended in Laemmli buffer, boiled for 10 min, and then analyzed by western blot.

## Co-immunoprecipitation experiments
For GFP-pulldown experiments HEK293T cells were seeded in 10 cm dishes and transfected 24 or 48 hours with 3 μg of the different EGFP construct expression plasmids. Transfected cells were washed three times with PBS before lysis in RIPA buffer (20 mM HEPES-HCl pH 7.4, 150 mM NaCl, 5 mM EDTA, 1% Triton X-100, 0.1% SDS, with protease inhibitors (ROCHE)) supplemented with ethidium bromide (400 μg/ml) when indicated. For the verification of protein interaction during infection we modified the previously established protocol[27]. Briefly, we transiently transfected HEK293-FcγRII cells with either EGFP or an EGFP-KAT7 fusion product. After 48 hours of transfection, the cells were washed and fresh DMEM with IPTG (1 mM) was added. *L. pneumophila* over-expressing either V5-LphD or V5-RomA were grown until post exponential phase in presence of 1 mM IPTG to allow fusion protein expression. After reaching an OD ~4, the bacteria were pre-opsonized by incubating them with an anti-FlaA antibody for 30 minutes at 37 °C. Then the cells are infected with MOI of 50. After one hour, the cells are washed and fresh DMEM (with 1 mM IPTG) is added. After 7 hours of infection, the cells are collected and lysed in RIPA buffer. To facilitate the lysis, cells were sonicated using a Bioruptor® Pico sonication device (Diagenode) for 15 cycles of 30 seconds ON/OFF. Lysates were precleared and the pulldown was performed using GFP-trap magnetic agarose beads (Chromotek), following the manufacturer's instructions at 4 °C overnight. Proteins were eluted in 30 μl Laemmli buffer and then analyzed by western blot or the beads directly processed for MS/MS analyses.

## Mass spectrometry analysis of GFP co-IP
MS grade Acetonitrile (ACN), MS grade H$_2$O and MS grade formic acid (FA) were acquired from Thermo Chemical. Proteins on magnetic beads were digested overnight at 37 °C with 1 μl (0.2 μg/μL) of trypsin (Promega) in a 25-mM NH$_4$HCO$_3$ buffer per sample. The resulting peptides were desalted using ZipTip μ-C18 Pipette Tips (Pierce Biotechnology). Samples were analyzed using an Orbitrap Fusion equipped with an easy spray ion source and coupled to a nano-LC Proxeon 1200 (Thermo Fisher Scientific). Peptides were loaded with an online preconcentration method and separated by chromatography using a Pepmap-RSLC C18 column (0.75 × 750 mm, 2 μm, 100 Å) from Thermo Fisher Scientific, equilibrated at 50 °C and operating at a flow rate of 300 nl/min. Peptides were eluted by a gradient of solvent A (H$_2$O, 0.1% FA) and solvent B (ACN/H$_2$O 80/20, 0.1% FA), the column was first equilibrated 5 min with 95% of A, then B was raised to 28% in 105 min and to 40% in 15 min. Finally, the column was washed with 95% B during 20 min and re-equilibrated at 95% A during 10 min. Peptide masses were analyzed in the Orbitrap cell in full ion scan mode, at a resolution of 120,000, a mass range of *m/z* 350-1550, and an AGC target of $4.10^5$. MS/MS were performed in the top speed 3 s mode. Peptides were selected for fragmentation by Higher-energy C-trap Dissociation (HCD) with a Normalized Collisional Energy of 27% and a dynamic exclusion of 60 s. Fragment masses were measured in an Ion trap in the rapid mode, with an AGC target of $1 \times 10^4$. Monocharged peptides and unassigned charge states were excluded from the MS/MS acquisition. The maximum ion accumulation times were set to 100 ms for MS and 35 ms for MS/MS acquisitions, respectively. Label-free quantification was done on Progenesis QI for Proteomics (Waters) in Hi-3 mode for protein abundance calculation. MGF peak files from Progenesis were processed by Proteome Discoverer 2.4 with the Sequest search engine. A custom database was created using the Swissprot/TrEMBL protein database release 2019_08 with the *Homo sapiens* taxonomy and including LphD, both from the *L. pneumophila* taxonomy. A maximum of 2 missed cleavages was authorized. Precursor and fragment mass tolerances were set to respectively 7 ppm and 0.5 Da. Spectra were filtered using a 1% FDR using the percolator node.

## In vitro protein binding assays
To assess protein-protein interactions in vitro, purified GST-RomA was incubated with equal amounts of HIS$_6$-LphD, HIS$_6$-LphD Y392F or HIS$_6$-LegK1 in protein binding buffer (25 mM Tris pH 8.0, 140 mM NaCl, 3 mM KCl, 0.1% NP40 with protease inhibitors (ROCHE)) overnight at 4 °C. GST-tagged proteins were pulled down using Glutathione agarose beads (Sigma). Beads were washed twice with protein binding buffer and proteins eluted using Laemmli buffer for immunoblot analyses. For the interaction of KAT7 with LphD and RomA the same approach was used. FLAG-KAT7 was incubated with either HIS$_6$-LphD or GST-RomA (HIS$_6$-LegK1 or GST-MLY1 as binding controls) and the pulldown was performed using DYKDDDDK Fab-Trap beads (Chromotek) followed by western blot analysis.

## Western blotting
Sample proteins were prepared in Laemmli sample buffer containing 400 mM β-mercaptoethanol and loaded on SDS PAGE gels, followed by a transfer onto a nitrocellulose membrane (0.2 μm; Trans-Blot Turbo system, Biorad). Ponceau S (SIGMA) staining was carried out to ensure equal protein loading. Membranes were blocked with 5% non-fat milk in TBS-Tween 0.5% for 1 hour and incubated with the respective primary antibody overnight at 4 °C. Antibodies used are listed in Table S3. Membranes were washed and probed with horseradish peroxidase-coupled antibody against either mouse IgG or rabbit IgG (1:2500 in 5% non-fat milk TBS-Tween) for 1 hour (Cell Signaling Technology). The proteins were visualized by chemiluminescence detection using HRP Substrate spray reagent (Advansta) on the G:BOX instrument (Syngene). Images were processed and quantified using Fiji software (ImageJ v2.9.0/1.53t)[42]. All uncropped and unprocessed scans of the western blots shown in the main and supplemental figures are provided in the Source Data file.

## RNA-sequencing
For the RNA-seq THP-1 monocytes were infected with either *L. pneumophila* wild-type, Δ*lphD*, Δ*romA* or Δ*lphD*-Δ*romA* strains, all of which were expressing EGFP under control of the *flaA* promoter. After

7 hours post-infection, the infected cells were enriched by FACS and stored at −80 °C. RNA-seq and data analysis was performed by Active Motif. In short, total RNA was isolated from samples using the Qiagen RNeasy Mini Kit (Qiagen). For each sample, 1 μg of total RNA was then used in the TruSeq Stranded mRNA Library kit (Illumina). Libraries were sequenced on Illumina NextSeq 500 as paired-end 42-nt reads. Sequence reads were analyzed with the STAR alignment v2.5.2b and the DeSeq2 package v1.14.1 was used to generate normalized counts[48].

### ChIP experiments
THP-1 monocytes were infected with either *L. pneumophila* wild type or Δ*lphD* strain, which both were expressing EGFP under control of the *flaA* promoter. After 7 h post-infection, the infected cells were enriched by FACS, crosslinked and stored at −80 °C until chromatin isolation. Chromatin immunoprecipitation (ChIP) experiments were undertaken as previously described[20]. DNA enrichment was followed by quantitative PCR (qPCR) (QuantStudio Thermo Fisher Scientific v2.6.0) with the primers listed in Table S2 and normalized using the Percent Input Method (ChIP signals divided by input sample signals).

### Production of anti-LphD antibodies in rabbit
Rabbit polyclonal antibodies to LphD were generated for this study (Thermo Fisher). Briefly, the purified His-LphD was injected in a rabbit for a protocol of 90 days. The produced antibody was qualitatively evaluated by affinity-purified ELISA: the concentration of the purified antibody is measured by indirect ELISA against the protein bound to the solid phase to determine the reactivity of the antibodies after elution.

### Statistical analyses
Graphs and statistical analysis were done in GraphPad Prism9 v9.5.0.

### Reporting summary
Further information on research design is available in the Nature Portfolio Reporting Summary linked to this article.

## Data availability
A reporting summary for this article is available as Supplementary Information file. The main data supporting the findings of this study are available within the article and its Supplementary Figures. The source data underlying figures and Supplementary Figures are provided as Source Data File. Additional details on datasets and protocols that support findings of this study will be made available by the corresponding authors upon reasonable request. The sequence reads as well as the coverage files of the RNAseq libraries of THP-1 cells generated in this study have been deposited in the NCBI Gene Expression Omnibus (GEO) database[49] under accession code GSE207487. To use Proteome Discoverer a custom database was created using the Swissprot/TrEMBL protein database release 2019_08 with the Homo sapiens taxonomy and including LphD, both from the *L. pneumophila* taxonomy. Source data are provided with this paper.

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

## Acknowledgements

Work in the C.B. laboratory is financed by the Institut Pasteur, the Fondation pour la Recherche Médicale (FRM) grant no. EQU201903007847, and the Agence Nationale de la Recherche grant no. ANR-10-LABX-62-IBEID to C.B. and grant no. ANR-18-CE15-0005-01 to M.R. We thank the members of the ProteoSeine@IJM facility (Institut Jacques Monod, CNRS UMR 7592 Paris University and the region Île-de-France) for mass spectrometry analyses. We thank Giulia Brenna for technical help and the group of Craig Roy for providing the HEK293-FcγRII cells. We thank Raphael Margueron for fruitful discussions. We gratefully acknowledge the UtechS Photonic BioImaging, C2RT, Institut Pasteur, supported by the French National Research Agency (France BioImaging; ANR-10–INSB-04; Investments for the Future) and the technical platform "BioProfiler-UFLC" (BFA Unit, Université Paris Cité) for provision of UFLC facilities. D.S. was funded by a Sorbonne University doctoral contract. C.D.S. is funded by an "Université de Paris Cité" doctoral contract.

## Author contributions

D.S., M.R., and C.B. designed research; D.S., M.R., S.M., J.B., C.D.S., and M.B.A. performed experiments; F.R.L. C.R. D.S. M.R. S.M. J.B. and A.W. analyzed data; and D.S., M.R., and C.B. wrote the paper.

## Competing interests

The authors declare no competing interests.
