## [Peer Review File · Nature Communications]

Legionella para-effectors target chromatin and promote bacterial replicationREVIEWER COMMENTS

Reviewer #1 (Remarks to the Author):

In this well-written manuscript, Schator and colleagues describe the discovery and characterisation of a histone deacetylase in *Legionella*, named LhpD. This enzyme specifically removes acetyl groups from H3K14 in the host nucleus. The authors propose this enzyme to work in concert with RomA, a previously described H3K14 methyltransferase that is similarly encoded by *Legionella*. These enzymes are suggested to function as para-effectors, working in coordination to modify host genomes and hijack the host response. The role for LhpD as H3K14 deacetylase is described convincingly, but questions remain on the functional impact on the host cell, and on how this enzyme cooperates with RomA.

I have the following comments:

Knockout of LhpD is essential for the work described in this manuscript. A more detailed description of the method used and of the validation should be added to the main manuscript.

Figure 2B: Please add single channel images to the manuscript. The level of H3K14ac shown in this figure seems incompatible with the modestly decreased level described in Figure 2D, which describes the more physiological effect. Why were cells in Figure 2C not stained for H3K14ac? Also, in the text I could not locate how long after transfection the cells in Figure 2B were fixed, nor if cells in used for Figure 2D were sorted for eGFP-positivity (rather than representing a mixture of infected and non-infected cells).

Figure 2D and 2G; Figure 2F and 2H: data showing the non-infected and delta-lphD in these panels look identical. Yet the legend describes 2 replicates for G and H, and 3 replicates for D and F. Both the duplication of the data (moreover without acknowledgement) and the lack of consistency in reporting the number of replicates are very troubling to me.

Figures 2I and 2J: why were only 2 replicates included here? I am also lacking negative controls in this analysis (to control for IP efficiency). Data presented in Figure 2J are used to confirm that LphD binds known RomA promoters, but none of these ChIPs seem to yield a significant signal.

Figures 3A and 3B describe *Legionella* growth curves, and are being interpreted as indicating that lphD is required for *Legionella* growth. To me however, the data suggest an initial delay, followed by a later normal growth (curves are parallel to one another with a similar slope, rather than of less steep slope as is to be expected if there is a genuine growth deficit).

Data presented in figures 3C-F are intriguing, and interpreted as indicating that LhpD and RomA are dependent on one another to promote bacterial replication. While this is one of the most intriguing observations in the manuscript, I could not find a convincing model explaining these data, nor data supporting mechanistic insights.

The data presented in Figure 4 provide an interesting glimpse into how LphD could reach its target sites. However, the IPs described cannot discriminate direct interactions from indirect, DNA-mediated interactions. Adding ethidium bromide to the IP buffers would be one way to mitigate such indirect interactions (cfr Lai & Herr, PNAS 1992).

Reviewer #2 (Remarks to the Author):

This study follows up on a very nice paper in *Cell Host & Microbe* by the authors showing that RomA methylates H3K14 and modulates host gene expression. However, H3K14 is usually acetylated, so it is unclear how RomA was able to access this residue for methylation. In this manuscript, the authors propose that lpp2163 is an effector with histone deacetylase activity that functions to facilitate the activity of the effector RomA. They coin the term 'para-effectors' to describe modulation of chromatin by LphD and RomA.

The ideas and data presented are quite interesting; however, the model that LphD and RomA

function as 'para-effectors' is premature. The authors should also include more details about how the data were analyzed, include additional controls, and more clearly indicate how many independent experiments were performed.

Major points:

The authors propose that LphD and RomA function sequentially, but there is no temporal analysis included in the manuscript. Are there any data to support earlier expression of LphD compared to RomA?

Figure 1H: It looks like total protein is decreased over time when octamers are incubated with active LphD. Densitometry should be performed to quantify relative H3K14ac. Total H3 should also be used as a loading here and elsewhere in the manuscript.

Figure 2A: The translocation data showing that LphD is an effector should be improved. It's unclear how the frequency of translocation was calculated and it's unclear to me why the data are presented this way as opposed to 460:535 ratios. Please also show representative wide-field micrographs. Although it's not necessary to include, have the authors observed translocation using any other standard assays (such as CyaA)?

Figure 2C: If possible, a clearer micrograph should be shown. Please also provide more information on infection conditions. What was the MOI? How long were the cells infected? Were the THP-1 cells differentiated with PMA? A negative control showing lack of signal when cells are infected with a dotA mutant would be beneficial. Do the authors detect less H3K14ac in the nucleus of cells infected with wildtype *L. pneumophila* compared to a Δ LphD mutant?

Figures 2D-H: It would be better to normalize these data to cells infected with the avirulent Δ dotA mutant. In the corresponding supplemental figures, H3K14ac abundance by western blot should also be normalized to total H3. It's unclear why H1 is used as opposed to H3.

Figure 2I: Since the Δ LphD strain is attenuated for replication, could differences be due to bacterial abundance?

Lines 205-209. Experiments using the Δ LphD Δ RomA strain are needed to draw the conclusion that RomA is responsible for the methylation observed.

Figure 3A-D. Please perform statistical analyses at each time point and indicate at which time points growth differs significantly.

Figure 4A. There appears to be less LphD Y392F bound to RomA compared to the wild-type. Is this phenotype consistent across experiments? A negative control should be included here. An unrelated HIS-tagged effector would be appropriate. Why is there a band - presumably corresponding to GST-MYL1 - in the LphD Y392F-GST-RomA pulldown? Please present the gels showing GST-fusions as one and include molecular weight markers.

Lines 282-284. Please show a representative Western blot in the supplement (like Fig S3F).

All experiments where chemical inhibitors are used (e.g. Fig 4J) should include a vehicle control (e.g. DMSO)

Minor points:

Line 409: Should say '..predicted structural similarity...'

Figure 1D and 1F should be removed since Figures 1G and 1H show the same thing and include genetic complementation.

Ensure that the number of independent experiments are clearly indicated in each figure legend.

Why isn't Figure S2G a main figure? It can take the place of Figure 3A.

Figure 3F can be moved to the supplemental information

Lines 310-312 and 324-326. Please present these data in the results section (can go in the supplement) or remove the statement.

Reviewer #3 (Remarks to the Author):

Schator et al. identify and characterize a novel *L. pneumophila* effector, LphD, that modifies the host chromatin to counteract the host defensive responses. They found that LphD is a HDAC that preferentially de-acetylates H3K14 and acts synergically with the pathogen methyl-transferase RomA that methylates H3K14. The authors define RomA and LphD as "Effector pair".

Overall, the manuscript is well written, the findings are novel and advance the understanding of how legionella hijacks host responses via chromatin modification, and the conclusions are well supported by the data. I am highly enthusiastic about the manuscript.

Minor comments:

- The manuscript reports "Methylation is generally associated to compaction of chromatin and reduced transcription" – this statement is not accurate, as histone methylation can be associated with both repression (H3K9me, H3K27me) and activation (H3K4me, H3K36me).
- The alpha fold prediction of H3K14 peptide interactions with LphD is not very compelling, due to the structural conformational flexibility of a short linear peptide. To strengthen this conclusions, other fitting models should be provided (for example another lysine of H3 or H4 tails). The limitation of alpha fold prediction with a short linear peptide should be mentioned in the text. Of note, LphD seems tractable for X-ray crystallography and obtain solid structural evidence of H3-LphD interactions. However, I understand this is beyond the scope of the paper and overall, it does not affect the major conclusions of the manuscript.
- I was not able to identify the buffer conditions for some enzymatic assays. For example, for the Fluor de Lys method in Fig 1G. Buffer conditions, should be as constant as possible across assays and clearly stated in methods or figure legend.
- Fig 1H: the manuscript should provide details on how the highly acetylated histone octamers are generated, and the buffer and salt conditions used in the assay. Histone octamers are stable at >250mM salt. If the reaction with LphD is performed at lower salt, this might result in octamer disassembly and formation of free histones Therefore the integrity of the octamers should be assessed to make conclusion about modification of the histone in the context of octamer.
- All enzymatic assays are done in the context of free histones or octamers (with concerns about octamer stability and histone proper folding). The most physiological relevant substrates are nucleosomes and experiments on nucleosome would be ideal and would also evaluate the effects of DNA, and histones beyond tails.
- Fig 4A: input should be shown
- Fig 4B: this experiment is not properly assessing the specific interaction with H3. The carryover of histones is often observed in this kind of assays. The authors should also show enrichment of the rest of the histones. A better way to assess the enrichment of LphD and RomA to chromatin, would be to perform nuclear fractionation and test the enrichment of LphD and RomA in the nuclear soluble, chromatin bound, and chromatin insoluble fractions.
- It is very intriguing that infection with double mutant LphD and RomA partially reverses the growth defect phenotypes. However, I do not understand the biological meaning of this results. Does it suggest that the cross-talk between H4K14 acetylation and methylation is not important for the replication of legionella? This is a very interesting observation but would require more investigation/explanation from the authors.

Other comments:

- Fig. 2E: H3K18ac goes up in infected cells, how do the authors explain this?
- Upon infection, changes in H3K14ac are milder than changes in H3K14me by blot. Do the authors have an explanation for this?
- Fig. 2I: H3K14me level are not down to NI levels in the delta-LphD. Do the authors have an explanation for this? Are other HDACs (from the host or pathogen) involved?

Reviewer #1 (Remarks to the Author):

In this well-written manuscript, Schator and colleagues describe the discovery and characterisation of a histone deacetylase in *Legionella*, named LhpD. This enzyme specifically removes acetyl groups from H3K14 in the host nucleus. The authors propose this enzyme to work in concert with RomA, a previously described H3K14 methyltransferase that is similarly encoded by *Legionella*. These enzymes are suggested to function as para-effectors, working in coordination to modify host genomes and hijack the host response. The role for LhpD as H3K14 deacetylase is described convincingly, but questions remain on the functional impact on the host cell, and on how this enzyme cooperates with RomA.

I have the following comments:

Knockout of LhpD is essential for the work described in this manuscript. A more detailed description of the method used and of the validation should be added to the main manuscript.

The detailed description of the method used for the construction was described in the Materials & Methods section that was provided in the supplement. However, as requested by the reviewer we have now moved the detailed description of the mutant construction to the M&M section of the manuscript and a short description was also added to the main text where the mutant was used the first time.

Lines 211-216 it reads now: "To analyse whether LphD has an impact on intracellular replication of *L. pneumophila* we constructed a $\Delta lphD$ mutant strain by replacing the chromosomal gene of the wild type strain by a gentamycin resistance cassette as previously described³⁷. Secondly, *lphD* was also replaced in the $\Delta romA$ strain to create a double mutant. All mutant strains were whole genome sequenced to ascertain the correct knock out of the gene and the absence of secondary mutations elsewhere."

Figure 2B: Please add single channel images to the manuscript.

As requested by the reviewer we have added single channel images for both GFP-LphD wt and GFP-LphD Y392F conditions (now **Figures 2B** and **S3C**).

The level of H3K14ac shown in this figure seems incompatible with the modestly decreased level described in Figure 2D, which describes the more physiological effect.

The level of H4K14ac observed in **Figure 2B** is highly decreased because of the transfection of the protein. In transfection conditions there is a high amount of protein in the cell, thus, the level of deacetylation is much higher than in "physiological" conditions, when the effector delivery is fine-tuned by the bacterium, and a much lower amount of protein is translocated than during transfection. Importantly, the deacetylation level of H3H14 is proportional to the amount of effector located in the nucleus. One can appreciate in **Figure 2D** that at the beginning of the infection (1-7 hours) the deacetylation progressively occurs, and it becomes complete at the end of the infectious cycle around 16 hours post-infection (**Figure 2D**).

Why were cells in **Figure 2C** not stained for H3K14ac?

As requested by the reviewer we have now also redone **Figure 2C** and have added staining for H3K14ac. It is clearly seen now that different to uninfected cells where the H4K14ac signal is high, H3K14ac is clearly decreased in cells where LphD is located in the nucleus.

Also, in the text I could not locate how long after transfection the cells in Figure 2B were fixed, nor if cells in used for Figure 2D were sorted for eGFP-positivity (rather than representing a mixture of infected and non-infected cells).

We apologize for this omission. We have added this information now to the Figure legend of Figure 2B and 2D.

Line 772-773 it reads now... HeLa cells were transfected with EGFP-LphD for 24hours and then stained for H3K14ac using a specific antibody.

Lines 782-783 and 787-788 it reads now...Cells were FACS sorted based on EGFP signal at the different times post-infection indicated to enrich infected cells.”

Figure 2D and 2G; Figure 2F and 2H: data showing the non-infected and delta-lphD in these panels look identical. Yet the legend describes 2 replicates for G and H, and 3 replicates for D and F. Both the duplication of the data (moreover without acknowledgement) and the lack of consistency in reporting the number of replicates are very troubling to me.

We apologize for this misunderstanding. The reviewer is correct, these figures represent in part the same data and we are aware of it. We presented it in this way as we thought a visual comparison of the wt - mutant experiment to the over expression would help the reader to better see the differences. However, as the reviewer saw it as data duplication, we realized that we had not made it clear. Indeed, when re-reading the figure legend, it was not explained. The independent experiments are independent experiments, but we should have clearly stated that for having a better visual comparison Figures 2F and 2H correspond to figures 2D and 2G in which we added the overexpression results. Both experiments were compared and normalized to non-infected cells. We apologize for this mistake. However, we propose to leave the over expression completely out as it is somewhat logic that if we overexpose a deacetylase that the acetylation goes more down than when we have only the native quantity. The results of the over expression do not change the message and the conclusion of the paper.

Figures 2I and 2J: why were only 2 replicates included here? I am also lacking negative controls in this analysis (to control for IP efficiency). Data presented in Figure 2J are used to confirm that LphD binds known RomA promoters, but none of these ChIPs seem to yield a significant signal.

As requested by the reviewer, we have repeated the ChIP experiments and have added the new replicates. Figure 2G (H3K14me) is now based on n=3 and Figure 2F (LphD) on n=4. Indeed, the ChIP does not give significant signals but a strong tendency for both the wild type and the $\Delta lphD$ mutant strain is seen. Furthermore, Figure 2G confirms our published data showing that RomA methylates H3K14 at promoter regions, and the results obtained for the $\Delta lphD$ mutant strain confirm the role of LphD for this methylation as observed at the general chromatin level (Figure 2F). Additionally Figure 2H shows that LphD indeed binds chromatin (further confirmed in Figure 4) and at the same promoter regions where RomA methylates H3K14. Thus, we think that these results together support our conclusions.

Figures 3A and 3B describe Legionella growth curves, and are being interpreted as indicating that lphD is required for Legionella growth. To me however, the data suggest an initial delay, followed by a later normal growth (curves are parallel to one another with a similar slope, rather than of less steep slope as is to be expected if there is a genuine growth deficit).

The reviewer might have overseen that the growth curves have been normalized to the input, which means that both, the wt and the mutant strain started with the same number of infecting bacteria. The mutant then shows a consistent growth delay compared to the wt strain. In addition, complementation reversed this growth delay showing that it is due to LphD. We also added the statistics and slope analyses in the figure legends. However, as well known in the *Legionella* field, there are only very few effectors that have an important phenotype in replication. To clarify this better we have changed the sentence as follows

Lines 219-222 it reads now... The $\Delta lphD$ strain showed a consistent growth delay compared to the wild type strain in both THP- 1 cells and in *A. castellanii* (**Figure 3A and 3B**). Complementation of the $\Delta lphD$ strain with full length *lphD* under the control of its native promotor completely reversed the growth delay

and even induced a slight increase in replication due to the plasmid copy number, further underlining the role of LphD in virulence of *L. pneumophila* (Figure 3C).

Data presented in figures 3C-F are intriguing, and interpreted as indicating that LhpD and RomA are dependent on one another to promote bacterial replication. While this is one of the most intriguing observations in the manuscript, I could not find a convincing model explaining these data, nor data supporting mechanistic insights.

The reviewer is correct, this is an intriguing phenotype and we have thus thought a lot about it and have searched the literature for such examples. We have found only one report of similar behavior of a double knockout strain. Liu and colleagues showed, that in *E. coli*, the knockout of the genes *relA* or *dnaK* causes a decrease in persister formation compared to wild type bacteria in the presence of Ampicillin, however, a double knockout ($\Delta relA \Delta dnaK$) showed a higher persister formation than the two single knockouts, closely resembling the wild type phenotype (Liu et al., 2017). The exact mechanism of this phenotype is also unknown, but it further highlights the complexity of genetic interactions within bacteria.

Thus, we hypothesise that the effect of the double knock out could be due to regulations within the bacterial cell, such as that the bacterial cell might compensate for the negative effects of the double knockouts by, for example, expressing other proteins involved in the intracellular cycle of *L. pneumophila* thereby counteracting the host response in a different way to allow effective replication. This might be possible because of the known redundancy of effector functions in the *Legionella* genomes, and the fact that for modulating one host pathway a number of different effectors acting on same and/or different steps of the same pathway are present.

Another hypothesis is that the absence of one of the two bacterial effectors, and the subsequent suboptimal modification of H3K14 triggers the host cell response, leading to an impaired bacterial replication. If both effectors are missing, this “incomplete” modification does not occur, hence there is no trigger for the host cell.

However, further experiments and analyses are necessary to completely understand the mechanism leading to this phenotype, thus we will pursue this question in further analyses.

Lines 347-364 it reads now... This is an intriguing finding and prompted us to search the literature for similar phenotypes. Indeed, Liu and colleagues showed, that in *Escherichia coli*, the knockout of the genes *relA* or *dnaK* causes a decrease in persister formation compared to wild type bacteria in the presence of ampicillin. However, a double knockout strain ($\Delta relA \Delta dnaK$) showed a higher persister formation than the two single knockouts, closely resembling the wild type phenotype (Liu et al., 2017). The exact mechanism of this phenotype is unknown.

We do not know either the exact mechanism but hypothesise that the phenotype of the $\Delta lphD \Delta romA$ double knock out strain could be due to specific regulations. In the double knock out the bacterial cell might compensate for the impact on replication by expressing other proteins involved in the intracellular cycle of *L. pneumophila* thereby counteracting the host response in a different way to allow effective replication. Indeed, it is known that the *Legionella* genomes encode for redundant effector functions. For modulating one host pathway a number of different effectors acting on same and/or different steps of the same pathway may be present (O'Connor TJ, Science. 2012 Dec 14;338(6113):1440-4.). Another possibility is that the absence of one of the two bacterial effectors, and the subsequent suboptimal modification of H3K14 triggers the host cell response, leading to an impaired bacterial replication. If both effectors are missing, this “incomplete” modification does not occur, hence there is no trigger for the host cell. However, further experiments and analyses are necessary to completely understand the mechanism leading to this phenotype.

The data presented in Figure 4 provide an interesting glimpse into how LphD could reach its target sites. However, the IPs described cannot discriminate direct interactions from indirect, DNA-mediated interactions. Adding ethidium bromide to the IP buffers would be one way to mitigate such indirect interactions (cfr Lai & Herr, PNAS 1992).

The binding assays we have presented in the manuscript, showing that LphD binds RomA (**Figure 4A**), and both bind KAT7 (**Figure 4H**) are *in vitro* binding assays that were performed in a DNA-free context. However, as requested by the reviewer, we now also performed an IP experiment in presence of ethidium bromide to exclude DNA-mediated interactions. Indeed, we observed the same interaction as in our *in vitro* binding assays; These new results are presented in **Figure S5G**.

Lines 279-284 it reads now... *In vitro* binding assays with tagged forms of LphD, RomA and KAT7 produced in *E. coli* showed that RomA and LphD bind KAT7 also in a chromatin-free context (**Figure 4H**). To exclude indirect interactions due to the presence of DNA we performed a GFP-trap to pull down EGFP-LphD and EGFP-RomA in the presence of ethidium bromide ⁽²⁸⁾. This confirmed together with the *in vitro* results that RomA and LphD interact with Kat7 (**Figure 4H**).

Reviewer #2 (Remarks to the Author):

This study follows up on a very nice paper in Cell Host & Microbe by the authors showing that RomA methylates H3K14 and modulates host gene expression. However, H3K14 is usually acetylated, so it is unclear how RomA was able to access this residue for methylation. In this manuscript, the authors propose that lpp2163 is an effector with histone deacetylase activity that functions to facilitate the activity of the effector RomA. They coin the term 'para-effectors' to describe modulation of chromatin by LphD and RomA.

The ideas and data presented are quite interesting; however, the model that LphD and RomA function as 'para-effectors' is premature. The authors should also include more details about how the data were analyzed, include additional controls, and more clearly indicate how many independent experiments were performed.

Major points:

The authors propose that LphD and RomA function sequentially, but there is no temporal analysis included in the manuscript. Are there any data to support earlier expression of lphD compared to romA?

Perhaps the reviewer has overseen it, but in **Figure 2D** and **F** a time course of the effects of RomA and LphD is presented, which shows opposing deacetylation and methylation dynamics at the functional level. Unfortunately, as the reviewer might know, we cannot measure finetuning of secretion. Furthermore, in **Figure 2F** it is shown that when LphD is missing there is much less methylation by RomA which again shows that the level of methylation is dependent on LphD activity. Indeed, methylation and acetylation on Lysine 14 are mutually exclusive on the epsilon-NH₂ group.

With respect to gene expression, the bacteria we used for infecting the THP-1 cells are at the post-exponential growth phase. At this phase we detect the expression of many secreted proteins that are described to function in the early stages of infection what led us to propose that these effectors need to be ready to be translocated upon the encounter of *Legionella* with a eukaryotic cell to win the battle against the host cell (Brüggemann et al., Cell Microbiol. (8):1228-40). Indeed, when analyzing the gene expression of LphD and RomA, we find LphD expressed in the late stages of *in vitro* growth or infection further supporting that it needs to be translocated very early in infection to act right away in the host cell. In contrast RomA is not expressed specifically in transmissive phase supporting the fact that it acts later in the life cycle. At steady state, HeLa and THP-1 cells have highly acetylated chromatin at Lys14. Indeed, as soon as 1h post infection with wt *Legionella* we start detecting deacetylation at Lys14 and methylation starts to increase.

Figure 1H: It looks like total protein is decreased over time when octamers are incubated with active LphD. Densitometry should be performed to quantify relative H3K14ac. Total H3 should also be used as a loading control here and elsewhere in the manuscript.

As requested by the reviewer we have added the densitometry analysis of the deacetylation assay of the octamers that was presented in **Figure 1H**. However, as reviewer 3 also asked for the nucleosome analyses, we have moved the blot that was shown in **Figure 1H** to the supplement (**now Figure S2B**) and have added also the densitometry analyses in the supplement (**now Figure S2A**) and have put the nucleosome analysis in the main Figure 1. As mentioned above as requested by reviewer#3 we confirmed these experiments on nucleosomes as substrate and these results are now presented in **Figure 1G** (densitometry analysis of deacetylase assays performed in triplicates on nucleosomes in which we used H3 as loading control) and a representative plot in **Figure S2C**.

We added these new results also in the text and the M&M section.

Lines 157-159 it reads now... Deacetylase activity of LphD on H3K14 was also observed when testing histone nucleosomes and octamers isolated from human cells (**Figure 1G, S2A, S2B, S2C**) with H3K14ac antibodies that we had validated by dot blot (**Figure S2D**).

Lines 556-571 it reads now...

In vitro nucleosome deacetylase activity assay

Assays were performed in a total volume of 20 μ L Tris 20 mM, 150 mM NaCl, pH 7.4, 2 mM DTT buffer containing 250 ng K14-acetylated recombinant mononucleosomes (#81001, Active motif) and 10 ng LphD (WT or Y392F mutant) at 30°C. At different time points (0, 1, 2, 3, 4 and 5 minutes), the reaction was stopped with the addition of 10 μ L Laemmli sample buffer containing 400 mM β -mercaptoethanol. Samples were analyzed on gradient 4-12% SDS PAGE, followed by a transfer onto a nitrocellulose membrane (0.2 μ m) at 210 mA for 65 min. Ponceau staining was carried out to ensure equal protein loading. Membranes were blocked with non-fat milk (5%) in PBS with 1% Tween (PBST) for 1 h and incubated either with α -H3K14ac antibody (1:10000, #ab52946 Abcam) in 1% non-fat milk PBST over night at 4°C. After washing 3 times, the membranes were incubated for 1 h at room temperature with peroxidase-coupled secondary antibody in 1% non-fat milk PBST. The proteins were then visualized by chemiluminescence detection using ECL reagent on LAS 4000 (Fujifilm) instrument. Membranes were finally stripped and reprobed with α -H3 antibody (1:10000, #3638 Cell Signaling) following the same protocol than before. Images were processed and quantified using ImageJ software (v1.46).

Lines 760-762 it reads now... Densitometry quantification of LphD activity on H3K14ac levels on nucleosomes. Nucleosomes were incubated with LphD (green) and LphD Y392F (black) and the reaction was stopped after the indicated time (in minutes).

With respect to the controls, we did not use histone H3 as control in all experiments performed here, but histone h1 instead, to have a loading control on the same western blot as the modified H3 proteins.

Figure 2A: The translocation data showing that LphD is an effector should be improved. It's unclear how the frequency of translocation was calculated and it's unclear to me why the data are presented this way as opposed to 460:535 ratios. Please also show representative wide-field micrographs. Although it's not necessary to include, have the authors observed translocation using any other standard assays (such as CyaA)?

The fluorescent ratio 460/535 is usually calculated when fluorescent microscopy images are taken after incubation of adherent infected cells with the beta-lactamase substrate. However, we did not use this technique in this way, but we have set up a cytofluorimetry approach to analyze effector translocation at single cell level (we counted around 20,000 cells per condition). The bar plot in **Figure 2A** represents the percentage of blue cells in which the fusion protein was translocated and the cleavage of CCF4 detected in the blue channel measured by flow cytometry. To better explain our approach, we have added in **Figure S3A** the FACS gating strategy and a representative FACS plot of each condition. Additionally, one can clearly observe translocation of LphD by immunofluorescence in **Figure 2C and S4B**.

Figure 2C: If possible, a clearer micrograph should be shown. Please also provide more information on infection conditions. What was the MOI? How long were the cells infected? Were the THP-1 cells differentiated with PMA? A negative control showing lack of signal when cells are infected with a *dotA* mutant would be beneficial. Do the authors detect less H3K14ac in the nucleus of cells infected with wildtype *L. pneumophila* compared to a Δ lphD mutant?

As requested, we added the infection conditions (MOI =50 and time of infection: 16 hours) that are given in the M& M section now also in the figure legend. We differentiated THP-1 with PMA in order to have adherent cells and be able to perform immunofluorescence experiments.

Lines 776-777 it reads now... Differentiated THP-1 cells were infected 16 hours at an MOI of 50 with *L. pneumophila* wild type expressing V5-LphD and GFP

We moved the original **Figure 2C**, showing the LphD nuclear localization by LphD staining to supplemental figure (**Figure S4B**), and replaced it as requested by the reviewer, with a new experiment. In the new **Figure 2C** we show the subcellular localization of LphD (V5-tagged), co-stained with H3K14ac. Due to antibody incompatibility, we cannot co-label LphD (rabbit) and H3K14ac (rabbit) in the same cell, thus we took a V5-labeled LphD. One can see in this figure that when LphD targets the nucleus of an infected cell, the H3K14ac mark importantly decreases. To note, we show here an image with both uninfected (NI) and infected (I) cells, in order to compare the level of H3K14ac between the uninfected cell (negative control) and the infected one. The uninfected cell can be considered equivalent to a cell infected with a Δ lphD mutant as seen in **Figure 2D**, the decrease of H3K14ac observed during *L. pneumophila* wild type infection is not observed when cells are infected with a Δ lphD mutant.

Since we already showed (**Figure 2A**), that LphD is secreted by the T4SS, we think it is irrelevant to show that in a Δ dotA background the effector is not reaching the nucleus (if the effector is not secreted in the cytosol, it cannot reach the nucleus neither).

Figures 2D-H: It would be better to normalize these data to cells infected with the avirulent Δ dotA mutant. In the corresponding supplemental figures, H3K14ac abundance by western blot should also be normalized to total H3. It's unclear why H1 is used as opposed to H3.

We did not use a Δ dotA strain here because, as Δ dotA is an avirulent strain, the infection rate is so low that is technically impossible to sort, and process, the same number of Δ dotA infected cells than wt or LphD mutant infected cells.

As mentioned above, we used H1 in order to have the staining of the control on the same membrane as the H3K14 modification, as it is more suitable for internal quantification of the WB signal.

Figure 2I: Since the Δ lphD strain is attenuated for replication, could differences be due to bacterial abundance?

As we measure the differences at 7h post infection, the difference cannot be due to different replication as the replication does merely start at this time point.

Lines 205-209 . Experiments using the Δ lphD Δ romA strain are needed to draw the conclusion that RomA is responsible for the methylation observed.

Nb: corresponds to fig 2J CHIP LphD

The reviewer might have overseen that we have published this result already in our Cell Host Microbe article Rolando. et al. 2013. There we have done CHIP with RomA and shown that it is responsible for H3K14 methylation (Rolando et al 2013). Thus, we cannot republish it here.

Figure 3A-D. Please perform statistical analyses at each time point and indicate at which time points

growth differs significantly.

As requested by the reviewer we added the statistical analyses in the figures and figure legends.

Lines 903-905 it reads now... Non-linear regression analysis using a straight-line model showed a significant difference in the slopes of the two conditions (P=0.0291).

Figure 4A. There appears to be less LphD Y392F bound to RomA compared to the wild-type. Is this phenotype consistent across experiments?

A negative control should be included here. An unrelated HIS-tagged effector would be appropriate.

Why is there a band - presumably corresponding to GST-MYL1 - in the LphD Y392F-GST-RomA pulldown?

Please present the gels showing GST-fusions as one and include molecular weight markers.

As requested by the reviewer we repeated the experiment by adding an unrelated His-tagged effector that is known to be secreted by the T4SS. We choose His-LegK1 (Lpp1439) (Ge et al., 2009; PNAS 106(33):13725-30), as negative control. One can see now in **Figure 4A** that LphD binds RomA *in vitro* but LegK1.

Lines 282-284. Please show a representative Western blot in the supplement (like Fig S3F).cit" This revealed that inhibition of KAT7 activity abolishes the decrease in H3K14 methylation observed in infection with *L. pneumophila* lacking LphD ($\Delta lphD$). Thus, LphD counteracts KAT7 activity (**Figure 4J**)."

As requested, we included a representative western blot in **Figure S5H**

Lines 290-291 it reads now... Thus, LphD counteracts KAT7 activity (**Figure 4J** and **Figure S5H**)

All experiments where chemical inhibitors are used (e.g. Fig 4J) should include a vehicle control (e.g. DMSO)

As requested, we have done these experiments also by adding as vehicle control EtOH because the inhibitor was dissolved in EtOH. The results of this experiments added below show that the result is not dependent on EtOH.

Quantification of western blot signal for H3K14 methylation in THP1 cells treated 7 hours with either EtOH (0.01% v/v) or 1 μ M MWM-3835 (KAT7-specific inhibitor). THP-1 cells were treated 18 hours prior to infection with *L. pneumophila* wild type or $\Delta lphD$ strain expressing EGFP. Histones were isolated from FACS-sorted cells and analysed by western blot. Histone H1 was used as loading control and signal is compared to non-infected cells EtOH treated (n = 3 \pm SEM).

Minor points:

Line 409: Should say '..predicted structural similarity...'

We corrected the title of the figure as requested.

Figure **1D** and **1F** should be removed since Figures 1G and 1H show the same thing and include genetic complementation.

We merged Figures 1E and 1F to have a unique **Figure 1E**. However, we did not remove it as reviewer#3 asked for having more understandable peptide predictions.

Ensure that the number of independent experiments are clearly indicated in each figure legend. We apologize if this was not done well enough. We have now clearly indicated the number of independent experiments in each Figure

Why isn't Figure S2G a main figure? It can take the place of Figure 3A. Figure 3F can be moved to the supplemental information

As requested, we moved the THP1 complementation to the main figure, now **Figure 3C**, and moved the RNAseq data in the Supplemental, now **Figure S5A**.

Lines 310-312 cit" Indeed, a putative NLS was predicted *in silico* at the N-terminus of LphD; however, its deletion did not affect nuclear translocation of LphD (data not shown)" and 324-326. "This is seen in transfection when high amounts of protein are delivered into a host cell, as LphD transfection in HeLa cells leads to the deacetylation of several other residues such as H3K18 and H3K23 (data not shown)". Please present these data in the results section (can go in the supplement) or remove the statement. As requested, we have added these data. The experiment where the predicted NLS was deleted is now shown in **Figure S6A**

Lines 317-319 it reads now... Indeed, a putative NLS was predicted *in silico* at the N-terminus of LphD; however, its deletion did not affect nuclear translocation of LphD (**Figure S6A**).

The results for the levels of H4K18ac and H3K23ac after transfection of LphD are now shown n **Figure S6B**

Lines 331-334 it reads now... This is seen in transfection when high amounts of protein are delivered into a host cell, as LphD transfection in HeLa cells leads to the deacetylation of several other residues such as H3K18 and H3K23 (**Figure S6B**).

Reviewer #3 (Remarks to the Author):

Schator et al. identify and characterize a novel *L. pneumophila* effector, LphD, that modifies the host chromatin to counteract the host defensive responses. They found that LphD is a HDAC that preferentially de-acetylates H3K14 and acts synergically with the pathogen methyl-transferase RomA that methylates H3K14. The authors define RomA and LphD as "Effector pair".

Overall, the manuscript is well written, the findings are novel and advance the understanding of how legionella hijacks host responses via chromatin modification, and the conclusions are well supported by the data. I am highly enthusiastic about the manuscript.

Minor comments:

- The manuscript reports "Methylation is generally associated to compaction of chromatin and reduced transcription" – this statement is not accurate, as histone methylation can be associated with both repression (H3K9me, H3K27me) and activation (H3K4me, H3K36me).

We apologize for this inaccuracy. We have changed this sentence.

Lines 73-77 it reads now... Methylation, depending on the target residue, can be associated with compaction of chromatin and reduced transcription⁵ whereas acetylation often impairs the affinity of

histones to DNA, consequently loosening chromatin compaction, promoting the recruitment of transcription factors ⁶ and increasing the mobility of histones along the DNA ⁷.

- The alpha fold prediction of H3K14 peptide interactions with LphD is not very compelling, due to the structural conformational flexibility of a short linear peptide. To strengthen this conclusions, other fitting models should be provided (for example another lysine of H3 or H4 tails). The limitation of alpha fold prediction with a short linear peptide should be mentioned in the text. Of note, LphD seems tractable for X-ray crystallography and obtain solid structural evidence of H3-LphD interactions. However, I understand this is beyond the scope of the paper and overall, it does not affect the major conclusions of the manuscript.

The reviewer may have overseen this part, but we discussed and showed the AlphaFold predictions for both H3 and H4, exactly for the reasons stated above (**Figures 1D, E, and Supplementary Figure S1C and S1D**).

Lines 139-150 it reads... Thus, we used AlphaFold to compute models of the complexes between LphD and the tails of histones H3 and H4 (residues 1-25) to see if we could get an indication for the physiological substrate specificity. In the case of H3, all 5 models show the same lysine (H3K14) binding to this pocket (**Figure S1C, S1D**). In contrast, for the H4 peptide the models were predicted with poor confidence and in only 2 out of 5 complexes is a lysine (H4K9) residue positioned at the active site (**Figure S1D, S1E**). Furthermore, whereas most of the H3 peptide is predicted with poor confidence and away from the LphD surface, H3K14 is placed into the active site pocket, a tight cavity that accommodates H3K14 and shows room for an additional acetyl group (**Figure 1D and 1E**). None of the other lysines of the peptide were predicted to bind LphD, probably due to specific substrate recognition through the flanking amino acids Pro12, Ala13 and Gly15, Gly16 (**Figure 1E inset**). Based on the results of these models, H3K14 seems to be the preferred target of LphD.

To further clarify this, we have now added a graphical representation (Supplementary **Figure S1E**) that shows the pLDDT confidence scores for each peptide residue. Only H3K14 and three surrounding residues are predicted confidently for all 5 models. For the H4 peptide, only 2 models predict H4K8 into the pocket, but the surrounding residues are predicted with low confidence, and 3 out of 5 do not give any confident prediction at all. We are confident that the AlphaFold predictions are correct, as confirmed by our *in vitro* work. We also conducted extensive crystallization trials with different constructs, and to date we could not obtain diffraction quality crystals. Further constructs may need to be tested but this is outside the scope of this work.

- I was not able to identify the buffer conditions for some enzymatic assays. For example, for the Fluor de Lys method in **Fig 1G**. Buffer conditions, should be as constant as possible across assays and clearly stated in methods or figure legend.

As requested, the reaction buffer used in the Fluor de Lys assay is now stated in the Materials and Methods section.

Lines 575-577 it reads now... "Briefly, different amount of purified LphD and catalytic inactive mutant LphD Y392F were incubated with the substrate in the reaction buffer (20 ml; 50mM TRIS/Cl, pH 8.0, 137mM NaCl, 2.7mM KCl, 1mM MgCl₂) for 30 minutes at 37°C."

- **Fig 1H**: the manuscript should provide details on how the highly acetylated histone octamers are generated, and the buffer and salt conditions used in the assay. Histone octamers are stable at >250mM salt. If the reaction with LphD is performed at lower salt, this might result in octamer disassembly and formation of free histones Therefore the integrity of the octamers should be assessed to make conclusion about modification of the histone in the context of octamer.

- All enzymatic assays are done in the context of free histones or octamers (with concerns about octamer stability and histone proper folding). The most physiological relevant substrates are nucleosomes and

experiments on nucleosome would be ideal and would also evaluate the effects of DNA, and histones beyond tails.

Highly acetylated histones were maintained in Tris 50mM, NaCl 50mM, pH8. This information was added to the M&M section.

Line 539 it reads now... "...(containing extracted histones) was buffer exchanged three times into Tris 50 mM, 50 mM NaCl, pH 8..."

As suggested by the reviewer, we confirmed the results obtained with octamers by performing deacetylase assays with nucleosomes as substrate. These new results are now added in **Figures 1G and S2C**. The results are consistent with those observed with octamers (now **Figure S2A and S2B**), confirming the deacetylase activity of LphD observed on H3K14.

Lines 157-159 it reads now... Deacetylase activity of LphD on H3K14 was also observed when testing histone nucleosomes or octamers isolated from human cells (**Figure 1G, S2A, S2B, S2C**) with H3K14ac antibodies that we had validated by dot blot (**Figure S2D**).

- **Fig 4A:** input should be shown

As requested by the reviewer we repeated the experiment by adding an unrelated His-tagged effector that is known to be secreted by the T4SS. We choose His-LegK1 (Lpp1439) (Ge et al., 2009; PNAS 106(33):13725-30), as negative control. One can see now in **Figure 4A** that LphD binds RomA *in vitro* but LegK1.

Additionally, we show now the input, as requested. In addition, we added in **Figure S5B** the Ponceau staining of the input membrane, to visualize the loaded proteins.

- **Fig 4B:** this experiment is not properly assessing the specific interaction with H3. The carryover of histones is often observed in this kind of assays. The authors should also show enrichment of the rest of the histones. A better way to assess the enrichment of LphD and RomA to chromatin, would be to perform nuclear fractionation and test the enrichment of LphD and RomA in the nuclear soluble, chromatin bound, and chromatin insoluble fractions.

As requested, we have performed nuclear fractionation of cells transfected with RomA or LphD and observed that both proteins are enriched in the nuclear fraction. These new results are added in **Figure S5C**

Lines 246-248 it reads now... Furthermore, we observed that when expressed in eukaryotic cells, RomA and LphD target the chromatin as they are enriched in the nuclear fraction (**Figure 4B, S4G**).

- It is very intriguing that infection with double mutant LphD and RomA partially reverses the growth defect phenotypes. However, I do not understand the biological meaning of this results. Does it suggest that the cross-talk between H4K14 acetylation and methylation is not important for the replication of legionella? This is a very interesting observation but would require more investigation/explanation from the authors.

The reviewer is correct, this is an intriguing phenotype and we have thus thought a lot about it and have searched the literature for such examples. We have found only one report of similar behavior of a double knockout strain. Liu and colleagues showed, that in *E. coli*, the knockout of the genes *relA* or *dnaK* causes a decrease in persister formation compared to wild type bacteria in the presence of Ampicillin, however, a double knockout ($\Delta relA \Delta dnaK$) showed a higher persister formation than the two single knockouts, closely resembling the wild type phenotype (Liu et al., 2017). The exact mechanism of this phenotype is also unknown, but it further highlights the complexity of genetic interactions within bacteria.

Thus, we hypothesise that the effect of the double knock out could be due to regulations within the bacterial cell, such as that the bacterial cell might compensate for the negative effects of the double knockouts by, for example, expressing other proteins involved in the intracellular cycle of *L. pneumophila* thereby counteracting the host response in a different way to allow effective replication. This might be possible because of the known redundancy of effector functions in the *Legionella* genomes, and the fact that for modulating one host pathway a number of different effectors acting on same and/or different steps of the same pathway are present.

Another hypothesis is that the absence of one of the two bacterial effectors, and the subsequent suboptimal modification of H3K14 triggers the host cell response, leading to an impaired bacterial replication. If both effectors are missing, this “incomplete” modification does not occur, hence there is no trigger for the host cell.

However, further experiments and analyses are necessary to completely understand the mechanism leading to this phenotype, thus we will pursue this question in further analyses.

Lines 347-364 it reads now... This is an intriguing finding and prompted us to search the literature for similar phenotypes. Indeed, Liu and colleagues showed, that in *Escherichia coli*, the knockout of the genes *relA* or *dnaK* causes a decrease in persister formation compared to wild type bacteria in the presence of ampicillin. However, a double knockout strain ($\Delta relA \Delta dnaK$) showed a higher persister formation than the two single knockouts, closely resembling the wild type phenotype (Liu et al., 2017). The exact mechanism of this phenotype is unknown.

We do not know either the exact mechanism but hypothesise that the phenotype of the $\Delta lphD \Delta romA$ double knock out strain could be due to specific regulations. In the double knock out the bacterial cell might compensate for the impact on replication by expressing other proteins involved in the intracellular cycle of *L. pneumophila* thereby counteracting the host response in a different way to allow effective replication. Indeed, it is known that the *Legionella* genomes encode for redundant effector functions. For modulating one host pathway a number of different effectors acting on same and/or different steps of the same pathway may be present (O'Connor TJ, Science. 2012 Dec 14;338(6113):1440-4.). Another possibility is that the absence of one of the two bacterial effectors, and the subsequent suboptimal modification of H3K14 triggers the host cell response, leading to an impaired bacterial replication. If both effectors are missing, this “incomplete” modification does not occur, hence there is no trigger for the host cell. However, further experiments and analyses are necessary to completely understand the mechanism leading to this phenotype

Other comments:

- Fig. 2E: H3K18ac goes up in infected cells, how do the authors explain this?

We apologize, but we do not understand what the reviewer asks for. H3K18ac is not increasing? Furthermore, also the Kd values obtained with the acetylated peptides do not increase (Figure 1F and Table 1)

- Upon infection, changes in H3K14ac are milder than changes in H3K14me by blot. Do the authors have an explanation for this?

K14ac is a dynamic modification that cells can control themselves (through a HAT/HDAC balance), K14me, however, is a more permanent mark (it seems the cell is not able to demethylate this residue), thus this is causing a strong accumulation of this mark and one can easily see by eye the accumulation, in contrast to K14Ac.

- Fig. 2I: H3K14me level are not down to NI levels in the delta-LphD. Do the authors have an explanation for this? Are other HDACs (from the host or pathogen) involved?

Indeed, we showed in Figure 2F that the levels of H3K14me are not completely reduced in absence of LphD meaning that the activity of RomA is partially due to the presence of LphD. One hypothesis is that other bacterial HDAC play a role in deacetylation H3K14 so RomA can methylate it. However, we could

not observe H3K14 deacetylation when cells are infected with the Δ lphD strains (**Figure 2D**). Thus, it is possible that RomA-dependent H3K14 methylation without LphD occurs “unspecifically” during the natural acetylation/deacetylation” turnover.

REVIEWERS' COMMENTS

Reviewer #1 (Remarks to the Author):

Thank you for providing a comprehensive set of answers to the comments I made on the previous version. These further solidify what was already a strong manuscript, and I have no further comments or questions.

Reviewer #2 (Remarks to the Author):

The authors have addressed most of my points with either a satisfactory explanation or incorporation of new data and the manuscript is much improved. However, some points could be more thoroughly addressed and these are shown in red text below the authors rebuttals in the attached file.

Reviewer #3 (Remarks to the Author):

The authors have addressed all the aspects raised. The manuscript is well written, the conclusions are supported by the data, and the findings are novel and advance the understanding of host-pathogen interactions. I am highly supportive of the publication of this manuscript.

Reviewer #2 (Remarks to the Author):

This study follows up on a very nice paper in Cell Host & Microbe by the authors showing that RomA methylates H3K14 and modulates host gene expression. However, H3K14 is usually acetylated, so it is unclear how RomA was able to access this residue for methylation. In this manuscript, the authors propose that Lpp2163 is an effector with histone deacetylase activity that functions to facilitate the activity of the effector RomA. They coin the term 'para-effectors' to describe modulation of chromatin by LphD and RomA.

The ideas and data presented are quite interesting; however, the model that LphD and RomA function as 'para-effectors' is premature. The authors should also include more details about how the data were analyzed, include additional controls, and more clearly indicate how many independent experiments were performed.

Major points:

The authors propose that LphD and RomA function sequentially, but there is no temporal analysis included in the manuscript. Are there any data to support earlier expression of LphD compared to RomA?

Perhaps the reviewer has overseen it, but in **Figure 2D** and **F** a time course of the effects of RomA and LphD is presented, which shows opposing deacetylation and methylation dynamics at the functional level. Unfortunately, as the reviewer might know, we cannot measure finetuning of secretion. Furthermore, in **Figure 2F** it is shown that when LphD is missing there is much less methylation by RomA which again shows that the level of methylation is dependent on LphD activity. Indeed, methylation and acetylation on Lysine 14 are mutually exclusive on the epsilon-NH₂ group.

With respect to gene expression, the bacteria we used for infecting the THP-1 cells are at the post-exponential growth phase. At this phase we detect the expression of many secreted proteins that are described to function in the early stages of infection what led us to propose that these effectors need to be ready to be translocated upon the encounter of *Legionella* with a eukaryotic cell to win the battle against the host cell (Brüggemann et al., Cell Microbiol. (8):1228-40). Indeed, when analyzing the gene expression of LphD and RomA, we find LphD expressed in the late stages of *in vitro* growth or infection further supporting that it needs to be translocated very early in infection to act right away in the host cell. In contrast RomA is not expressed specifically in transmissive phase supporting the fact that it acts later in the life cycle. At steady state, HeLa and THP-1 cells have highly acetylated chromatin at Lys14. Indeed, as soon as 1h post infection with wt *Legionella* we start detecting deacetylation at Lys14 and methylation starts to increase.

Figure 1H: It looks like total protein is decreased over time when octamers are incubated with active LphD. Densitometry should be performed to quantify relative H3K14ac. Total H3 should also be used as a loading control here and elsewhere in the manuscript.

As requested by the reviewer we have added the densitometry analysis of the deacetylation assay of the octamers that was presented in **Figure 1H**. However, as reviewer 3 also asked for the nucleosome analyses, we have moved the blot that was shown in **Figure 1H** to the supplement (**now Figure S2B**) and have added also the densitometry analyses in the supplement (**now Figure S2A**) and have put the nucleosome analysis in the main Figure 1. As mentioned above as requested by reviewer#3 we confirmed these experiments on nucleosomes as substrate and these results are now presented in **Figure 1G** (densitometry analysis of deacetylase assays performed in triplicates on nucleosomes in which we used H3 as loading control) and a representative plot in **Figure S2C**.

We added these new results also in the text and the M&M section.

Lines 157-159 it reads now... Deacetylase activity of LphD on H3K14 was also observed when testing histone nucleosomes and octamers isolated from human cells (**Figure 1G, S2A, S2B, S2C**) with H3K14ac antibodies that we had validated by dot blot (**Figure S2D**).

Lines 556-571 it reads now...

In vitro nucleosome deacetylase activity assay

Assays were performed in a total volume of 20 μ L Tris 20 mM, 150 mM NaCl, pH 7.4, 2 mM DTT buffer containing 250 ng K14-acetylated recombinant mononucleosomes (#81001, Active motif) and 10 ng LphD (WT or Y392F mutant) at 30°C. At different time points (0, 1, 2, 3, 4 and 5 minutes), the reaction was stopped with the addition of 10 μ L Laemmli sample buffer containing 400 mM β -mercaptoethanol. Samples were analyzed on gradient 4-12% SDS PAGE, followed by a transfer onto a nitrocellulose membrane (0.2 μ m) at 210 mA for 65 min. Ponceau staining was carried out to ensure equal protein loading. Membranes were blocked with non-fat milk (5%) in PBS with 1% Tween (PBST) for 1 h and incubated either with α -H3K14Ac antibody (1:10000, #ab52946 Abcam) in 1% non-fat milk PBST over night at 4°C. After washing 3 times, the membranes were incubated for 1 h at room temperature with peroxidase-coupled secondary antibody in 1% non-fat milk PBST. The proteins were then visualized by chemiluminescence detection using ECL reagent on LAS 4000 (Fujifilm) instrument. Membranes were finally stripped and reprobed with α -H3 antibody (1:10000, #3638 Cell Signaling) following the same protocol than before. Images were processed and quantified using ImageJ software (v1.46).

Lines 760-762 it reads now... Densitometry quantification of LphD activity on H3K14ac levels on nucleosomes. Nucleosomes were incubated with LphD (green) and LphD Y392F (black) and the reaction was stopped after the indicated time (in minutes).

With respect to the controls, we did not use histone H3 as control in all experiments performed here, but histone h1 instead, to have a loading control on the same western blot as the modified H3 proteins.

Figure 2A: The translocation data showing that LphD is an effector should be improved. It's unclear how the frequency of translocation was calculated and it's unclear to me why the data are presented this way as opposed to 460:535 ratios. Please also show representative wide-field micrographs. Although it's not necessary to include, have the authors observed translocation using any other standard assays (such as CyaA)?

The fluorescent ratio 460/535 is usually calculated when fluorescent microscopy images are taken after incubation of adherent infected cells with the beta-lactamase substrate. However, we did not use this technique in this way, but we have set up a cytofluorimetry approach to analyze effector translocation at single cell level (we counted around 20,000 cells per condition). The bar plot in **Figure 2A** represents the percentage of blue cells in which the fusion protein was translocated and the cleavage of CCF4 detected in the blue channel measured by flow cytometry. To better explain our approach, we have added in **Figure S3A** the FACS gating strategy and a representative FACS plot of each condition. Additionally, one can clearly observe translocation of LphD by immunofluorescence in **Figure 2C and S4B**.

Figure 2C: If possible, a clearer micrograph should be shown. Please also provide more information on infection conditions. What was the MOI? How long were the cells infected? Were the THP-1 cells differentiated with PMA? A negative control showing lack of signal when cells are infected with a dotA mutant would be beneficial. Do the authors detect less H3K14ac in the nucleus of cells infected with wildtype *L. pneumophila* compared to a Δ LphD mutant?

As requested, we added the infection conditions (MOI =50 and time of infection: 16 hours) that are given in the M& M section now also in the figure legend. We differentiated THP-1 with PMA in order to have adherent cells and be able to perform immunofluorescence experiments.

Lines 776-777 it reads now... Differentiated THP-1 cells were infected 16 hours at an MOI of 50 with *L. pneumophila* wild type expressing V5-LphD and GFP

We moved the original **Figure 2C**, showing the LphD nuclear localization by LphD staining to supplemental figure (**Figure S4B**), and replaced it as requested by the reviewer, with a new experiment. In the new **Figure 2C** we show the subcellular localization of LphD (V5-tagged), co-stained with H3K14ac. Due to antibody incompatibility, we cannot co-label LphD (rabbit) and H3K14ac (rabbit) in the same cell, thus we took a V5-labeled LphD. One can see in this figure that when LphD targets the nucleus of an infected cell, the H3K14ac mark importantly decreases. To note, we show here an image with both uninfected (NI) and infected (I) cells, in order to compare the level of H3K14ac between the uninfected cell (negative control) and the infected one. The uninfected cell can be considered equivalent to a cell infected with a Δ *lphD* mutant as seen in **Figure 2D**, the decrease of H3K14ac observed during *L. pneumophila* wild type infection is not observed when cells are infected with a Δ *lphD* mutant.

Fig 2C : The authors state that the images presented in Fig 2C are sufficient to show differences in H3K14ac abundance within infected and uninfected cells can be observed image shown. The image shown is convincing ; however, a single micrograph is not sufficient to draw conclusions. A callout to Figure 2C also seems to be missing in the text of the revised manuscript.

I also stand by my ascertainment that macrophages infected with avirulent *L. pneumophila* would be a more appropriate control than non-infected cells. The authors claim that the data in Fig 2D are sufficient to show that uninfected cells are equivalent to cells infected with *L. pneumophila* Δ *lphD* since there were differences in H3K14ac signal by Western blot compared to WT infection. However, as I'm sure the authors are aware, macrophage gene expression changes drastically when TLR signaling is induced by either virulent and avirulent bacteria. Thus, non-infected cells are never really "equivalent" to those that are exposed to bacteria. Despite the added rigor that would be afforded by leveraging avirulent *L. pneumophila* as a control, I acknowledge that these data are not crucial for this study.

Since we already showed (**Figure 2A**), that LphD is secreted by the T4SS, we think it is irrelevant to show that in a Δ *dotA* background the effector is not reaching the nucleus (if the effector is not secreted in the cytosol, it cannot reach the nucleus neither).

I apologize for the confusion, I meant to suggest evaluating whether changes in H3K14ac are induced by bacterial infection alone by quantifying changes within cells infected with the Δ *dotA* strain normalized to uninfected cells (by either Western blot or microscopy). While these data would add rigor to the study, they are not necessary to support the main conclusions.

Figures 2D-H: It would be better to normalize these data to cells infected with the avirulent Δ *dotA* mutant. In the corresponding supplemental figures, H3K14ac abundance by western blot should also be normalized to total H3. It's unclear why H1 is used as opposed to H3.

We did not use a Δ *dotA* strain here because, as Δ *dotA* is an avirulent strain, the infection rate is so low that is technically impossible to sort, and process, the same number of Δ *dotA* infected cells than wt or LphD mutant infected cells.

Have the authors seen this experimentally? I am unaware of any study showing that avirulent *L. pneumophila* are phagocytosed less effectively than virulent bacteria and it is also clear that avirulent bacteria retain some viability within macrophages even though they are not actively replicating. So, with such a high MOI, I'm surprised that GFP signal would be so low as to prevent effective sorting. Again, this control would improve the rigor of that manuscript but are not essential to support the overarching conclusions.

As mentioned above, we used H1 in order to have the staining of the control on the same membrane as the H3K14 modification, as it is more suitable for internal quantification of the WB signal.

Figure 2I: Since the Δ lphD strain is attenuated for replication, could differences be due to bacterial abundance?

As we measure the differences at 7h post infection, the difference cannot be due to different replication as the replication does merely start at this time point.

This is true, but differences in bacterial abundance can be mediated by differences in phagocytosis and it is unclear whether this is the case since all intracellular replication data are shown as fold changes in CFU normalized to 2h post-infection for each strain. The authors could consider presenting growth data as CFU/mL or showing CFU counts after 2h of infection.

Lines 205-209 . Experiments using the Δ lphD Δ romA strain are needed to draw the conclusion that RomA is responsible for the methylation observed.

Nb: corresponds to fig 2J CHIP LphD

The reviewer might have overseen that we have published this result already in our Cell Host Microbe article Rolando. et al. 2013. There we have done CHIP with RomA and shown that it is responsible for H3K14 methylation (Rolando et al 2013). Thus, we cannot republish it here.

I agree that the previous report nicely shows that RomA methylates H3K14; however, data from a previous publication cannot serve as a control in the present study. Results from previous studies are often confirmed as controls in manuscripts, so I am confused about perceived issues 'republishing' the RomA phenotype. Without data showing that methylation is abolished by additional loss of *romA* from LphD-deficient bacteria, causation can merely be hypothesized and the text in lines 203-208 should be revised accordingly.

Figure 3A-D. Please perform statistical analyses at each time point and indicate at which time points growth differs significantly.

As requested by the reviewer we added the statistical analyses in the figures and figure legends.

Lines 903-905 it reads now... Non-linear regression analysis using a straight-line model showed a significant difference in the slopes of the two conditions (P=0.0291).

Figure 4A. There appears to be less LphD Y392F bound to RomA compared to the wild-type. Is this phenotype consistent across experiments?

A negative control should be included here. An unrelated HIS-tagged effector would be appropriate. Why is there a band - presumably corresponding to GST-MYL1 - in the LphD Y392F-GST-RomA pulldown? Please present the gels showing GST-fusions as one and include molecular weight markers.

As requested by the reviewer we repeated the experiment by adding an unrelated His-tagged effector that is known to be secreted by the T4SS. We choose His-LegK1 (Lpp1439) (Ge et al., 2009; PNAS 106(33):13725-30), as negative control. One can see now in **Figure 4A** that LphD binds RomA *in vitro* but LegK1.

Lines 282-284. Please show a representative Western blot in the supplement (like Fig S3F).cit" This revealed that inhibition of KAT7 activity abolishes the decrease in H3K14 methylation observed in infection with *L. pneumophila* lacking LphD (Δ lphD). Thus, LphD counteracts KAT7 activity (**Figure 4J**)."

As requested, we included a representative western blot in **Figure S5H**

Lines 290-291 it reads now... Thus, LphD counteracts KAT7 activity (**Figure 4J** and **Figure S5H**)

All experiments where chemical inhibitors are used (e.g. Fig 4J) should include a vehicle control (e.g. DMSO)

As requested, we have done these experiments also by adding as vehicle control EtOH because the inhibitor was dissolved in EtOH. The results of this experiments added below show that the result is not dependent on EtOH.

Quantification of western blot signal for H3K14 methylation in THP1 cells treated 7 hours with either EtOH (0.01% v/v) or 1 μ M MWM-3835 (KAT7-specific inhibitor). THP-1 cells were treated 18 hours prior to infection with *L. pneumophila* wild type or Δ lphD strain expressing EGFP. Histones were isolated from FACS-sorted cells and analysed by western blot. Histone H1 was used as loading control and signal is compared to non-infected cells EtOH treated (n = 3 \pm SEM).

Minor points:

Line 409: Should say '..predicted structural similarity...'
We corrected the title of the figure as requested.

Figure **1D** and **1F** should be removed since Figures 1G and 1H show the same thing and include genetic complementation.

We merged Figures 1E and 1F to have a unique **Figure 1E**. However, we did not remove it as reviewer#3 asked for having more understandable peptide predictions.

Ensure that the number of independent experiments are clearly indicated in each figure legend.
We apologize if this was not done well enough. We have now clearly indicated the number of independent experiments in each Figure

Why isn't Figure S2G a main figure? It can take the place of Figure 3A. Figure 3F can be moved to the supplemental information

As requested, we moved the THP1 complementation to the main figure, now **Figure 3C**, and moved the RNAseq data in the Supplemental, now **Figure S5A**.

Lines 310-312 cit' Indeed, a putative NLS was predicted *in silico* at the N-terminus of LphD; however, its deletion did not affect nuclear translocation of LphD (data not shown)" and 324-326. "This is seen in transfection when high amounts of protein are delivered into a host cell, as LphD transfection in HeLa cells leads to the deacetylation of several other residues such as H3K18 and H3K23 (data not shown)". Please present these data in the results section (can go in the supplement) or remove the statement.

As requested, we have added these data. The experiment where the predicted NLS was deleted is now shown in **Figure S6A**

Lines 317-319 it reads now... Indeed, a putative NLS was predicted *in silico* at the N-terminus of LphD; however, its deletion did not affect nuclear translocation of LphD (**Figure S6A**).

The results for the levels of H4K18ac and H3K23ac after transfection of LphD are now shown n **Figure S6B**

Lines 331-334 it reads now... This is seen in transfection when high amounts of protein are delivered into a host cell, as LphD transfection in HeLa cells leads to the deacetylation of several other residues such as H3K18 and H3K23 (**Figure S6B**).

Reviewer #2 (Remarks to the Author):

This study follows up on a very nice paper in Cell Host & Microbe by the authors showing that RomA methylates H3K14 and modulates host gene expression. However, H3K14 is usually acetylated, so it is unclear how RomA was able to access this residue for methylation. In this manuscript, the authors propose that Lpp2163 is an effector with histone deacetylase activity that functions to facilitate the activity of the effector RomA. They coin the term 'para-effectors' to describe modulation of chromatin by LphD and RomA.

The ideas and data presented are quite interesting; however, the model that LphD and RomA function as 'para-effectors' is premature. The authors should also include more details about how the data were analyzed, include additional controls, and more clearly indicate how many independent experiments were performed.

Major points:

The authors propose that LphD and RomA function sequentially, but there is no temporal analysis included in the manuscript. Are there any data to support earlier expression of LphD compared to RomA?

Perhaps the reviewer has overseen it, but in **Figure 2D** and **F** a time course of the effects of RomA and LphD is presented, which shows opposing deacetylation and methylation dynamics at the functional level. Unfortunately, as the reviewer might know, we cannot measure finetuning of secretion. Furthermore, in **Figure 2F** it is shown that when LphD is missing there is much less methylation by RomA which again shows that the level of methylation is dependent on LphD activity. Indeed, methylation and acetylation on Lysine 14 are mutually exclusive on the epsilon-NH₂ group.

With respect to gene expression, the bacteria we used for infecting the THP-1 cells are at the post-exponential growth phase. At this phase we detect the expression of many secreted proteins that are described to function in the early stages of infection what led us to propose that these effectors need to be ready to be translocated upon the encounter of *Legionella* with a eukaryotic cell to win the battle against the host cell (Brüggemann et al., Cell Microbiol. (8):1228-40). Indeed, when analyzing the gene expression of LphD and RomA, we find LphD expressed in the late stages of *in vitro* growth or infection further supporting that it needs to be translocated very early in infection to act right away in the host cell. In contrast RomA is not expressed specifically in transmissive phase supporting the fact that it acts later in the life cycle. At steady state, HeLa and THP-1 cells have highly acetylated chromatin at Lys14. Indeed, as soon as 1h post infection with wt *Legionella* we start detecting deacetylation at Lys14 and methylation starts to increase.

Figure 1H: It looks like total protein is decreased over time when octamers are incubated with active LphD. Densitometry should be performed to quantify relative H3K14ac. Total H3 should also be used as a loading control here and elsewhere in the manuscript.

As requested by the reviewer we have added the densitometry analysis of the deacetylation assay of the octamers that was presented in **Figure 1H**. However, as reviewer 3 also asked for the nucleosome analyses, we have moved the blot that was shown in **Figure 1H** to the supplement (**now Figure S2B**) and have added also the densitometry analyses in the supplement (**now Figure S2A**) and have put the nucleosome analysis in the main Figure 1. As mentioned above as requested by reviewer#3 we confirmed these experiments on nucleosomes as substrate and these results are now presented in **Figure 1G** (densitometry analysis of deacetylase assays performed in triplicates on nucleosomes in which we used H3 as loading control) and a representative plot in **Figure S2C**.

We added these new results also in the text and the M&M section.

Lines 157-159 it reads now... Deacetylase activity of LphD on H3K14 was also observed when testing histone nucleosomes and octamers isolated from human cells (**Figure 1G, S2A, S2B, S2C**) with H3K14ac antibodies that we had validated by dot blot (**Figure S2D**).

Lines 556-571 it reads now...

In vitro nucleosome deacetylase activity assay

Assays were performed in a total volume of 20 μ L Tris 20 mM, 150 mM NaCl, pH 7.4, 2 mM DTT buffer containing 250 ng K14-acetylated recombinant mononucleosomes (#81001, Active motif) and 10 ng LphD (WT or Y392F mutant) at 30°C. At different time points (0, 1, 2, 3, 4 and 5 minutes), the reaction was stopped with the addition of 10 μ L Laemmli sample buffer containing 400 mM β -mercaptoethanol. Samples were analyzed on gradient 4-12% SDS PAGE, followed by a transfer onto a nitrocellulose membrane (0.2 μ m) at 210 mA for 65 min. Ponceau staining was carried out to ensure equal protein loading. Membranes were blocked with non-fat milk (5%) in PBS with 1% Tween (PBST) for 1 h and incubated either with α -H3K14Ac antibody (1:10000, #ab52946 Abcam) in 1% non-fat milk PBST over night at 4°C. After washing 3 times, the membranes were incubated for 1 h at room temperature with peroxidase-coupled secondary antibody in 1% non-fat milk PBST. The proteins were then visualized by chemiluminescence detection using ECL reagent on LAS 4000 (Fujifilm) instrument. Membranes were finally stripped and reprobed with α -H3 antibody (1:10000, #3638 Cell Signaling) following the same protocol than before. Images were processed and quantified using ImageJ software (v1.46).

Lines 760-762 it reads now... Densitometry quantification of LphD activity on H3K14ac levels on nucleosomes. Nucleosomes were incubated with LphD (green) and LphD Y392F (black) and the reaction was stopped after the indicated time (in minutes).

With respect to the controls, we did not use histone H3 as control in all experiments performed here, but histone h1 instead, to have a loading control on the same western blot as the modified H3 proteins.

Figure 2A: The translocation data showing that LphD is an effector should be improved. It's unclear how the frequency of translocation was calculated and it's unclear to me why the data are presented this way as opposed to 460:535 ratios. Please also show representative wide-field micrographs. Although it's not necessary to include, have the authors observed translocation using any other standard assays (such as CyaA)?

The fluorescent ratio 460/535 is usually calculated when fluorescent microscopy images are taken after incubation of adherent infected cells with the beta-lactamase substrate. However, we did not use this technique in this way, but we have set up a cytofluorimetry approach to analyze effector translocation at single cell level (we counted around 20,000 cells per condition). The bar plot in **Figure 2A** represents the percentage of blue cells in which the fusion protein was translocated and the cleavage of CCF4 detected in the blue channel measured by flow cytometry. To better explain our approach, we have added in **Figure S3A** the FACS gating strategy and a representative FACS plot of each condition. Additionally, one can clearly observe translocation of LphD by immunofluorescence in **Figure 2C and S4B**.

Figure 2C: If possible, a clearer micrograph should be shown. Please also provide more information on infection conditions. What was the MOI? How long were the cells infected? Were the THP-1 cells differentiated with PMA? A negative control showing lack of signal when cells are infected with a dotA mutant would be beneficial. Do the authors detect less H3K14ac in the nucleus of cells infected with wildtype *L. pneumophila* compared to a Δ lphD mutant?

As requested, we added the infection conditions (MOI =50 and time of infection: 16 hours) that are given in the M& M section now also in the figure legend. We differentiated THP-1 with PMA in order to have adherent cells and be able to perform immunofluorescence experiments.

Lines 776-777 it reads now... Differentiated THP-1 cells were infected 16 hours at an MOI of 50 with *L. pneumophila* wild type expressing V5-LphD and GFP

We moved the original **Figure 2C**, showing the LphD nuclear localization by LphD staining to supplemental figure (**Figure S4B**), and replaced it as requested by the reviewer, with a new experiment. In the new **Figure 2C** we show the subcellular localization of LphD (V5-tagged), co-stained with H3K14ac. Due to antibody incompatibility, we cannot co-label LphD (rabbit) and H3K14ac (rabbit) in the same cell, thus we took a V5-labeled LphD. One can see in this figure that when LphD targets the nucleus of an infected cell, the H3K14ac mark importantly decreases. To note, we show here an image with both uninfected (NI) and infected (I) cells, in order to compare the level of H3K14ac between the uninfected cell (negative control) and the infected one. The uninfected cell can be considered equivalent to a cell infected with a Δ *lphD* mutant as seen in **Figure 2D**, the decrease of H3K14ac observed during *L. pneumophila* wild type infection is not observed when cells are infected with a Δ *lphD* mutant.

Fig 2C : The authors state that the images presented in Fig 2C are sufficient to show differences in H3K14ac abundance within infected and uninfected cells can be observed image shown. The image shown is convincing ; however, a single micrograph is not sufficient to draw conclusions.

Answer:

We agree with the reviewer, but the single micrograph is not our result but was only included to support the WB quantification experiments (N=3) showing the global decrease in of H4K14ac observed upon infection with Lpp wt strain (Fig2D).

A callout to Figure 2C also seems to be missing in the text of the revised manuscript.

Answer:

Thank you for pointing out this omission. We corrected the callout to “Figure2D” line 183 to “Figure 2C” line 184 in the text.

I also stand by my ascertainment that macrophages infected with avirulent *L. pneumophila* would be a more appropriate control than non-infected cells. The authors claim that the data in Fig 2D are sufficient to show that uninfected cells are equivalent to cells infected with *L. pneumophila* Δ *lphD* since there were differences in H3K14ac signal by Western blot compared to WT infection. However, as I’m sure the authors are aware, macrophage gene expression changes drastically when TLR signaling is induced by either virulent and avirulent bacteria. Thus, non-infected cells are never really “equivalent” to those that are exposed to bacteria. Despite the added rigor that would be afforded by leveraging avirulent *L. pneumophila* as a control, I acknowledge that these data are not crucial for this study.

Answer:

We are aware that macrophage gene expression changes when TLR signaling is induced and that non-infected cells are not completely “equivalent” to bacteria exposed cells. However, as the point of figure 2D was to determine the direct role of LphD in targeting Lys14 of H3 we compared the degree of this modification to non-infected cells (with the final objective to compare wt and Δ *lphD*). Of course, we do not exclude that in addition to LphD, another *Legionella* effector secreted by the T4SS may directly or indirectly decrease H3K14Ac. This point could be investigated by comparing levels of H3K14ac in non-infected vs Δ *dotA* vs Δ *lphD* strains, but this is not the question of this work.

Since we already showed (**Figure 2A**), that LphD is secreted by the T4SS, we think it is irrelevant to show that in a Δ *dotA* background the effector is not reaching the nucleus (if the effector is not secreted in the cytosol, it cannot reach the nucleus neither).

I apologize for the confusion, I meant to suggest evaluating whether changes in H3K14ac are induced by bacterial infection alone by quantifying changes within cells infected with the Δ *dotA* strain normalized to uninfected cells (by either Western blot or microscopy). While these data would add rigor to the study, they are not necessary to support the main conclusions.

Answer:

We do not exclude that bacterial infection alone could induce changes in H3K14Ac. As answered before, avirulent $\Delta dotA$ infection could induce indirect (via TLR activation) changes in H3K14Ac, if compared to both uninfected and wt. However, as already stated below, this was not the aim of Figure 2 (in particular 2C-D) where we wanted to show a direct role of LphD in targeting H3K14 during infection.

Figures 2D-H: It would be better to normalize these data to cells infected with the avirulent $\Delta dotA$ mutant. In the corresponding supplemental figures, H3K14ac abundance by western blot should also be normalized to total H3. It's unclear why H1 is used as opposed to H3.

We did not use a $\Delta dotA$ strain here because, as $\Delta dotA$ is an avirulent strain, the infection rate is so low that is technically impossible to sort, and process, the same number of $\Delta dotA$ infected cells than wt or LphD mutant infected cells.

Have the authors seen this experimentally? I am unaware of any study showing that avirulent *L. pneumophila* are phagocytosed less effectively than virulent bacteria and it is also clear that avirulent bacteria retain some viability within macrophages even though they are not actively replicating. So, with such a high MOI, I'm surprised that GFP signal would be so low as to prevent effective sorting. Again, this control would improve the rigor of that manuscript but are not essential to support the overarching conclusions.

Answer:

Yes, we have seen this, as in all our experiments the $\Delta dotA$ strain is less infective than the virulent strain. As suggested by the reviewer and already discussed above, Figure 2 aimed in confirming LphD activity in infected cells and thus adding a $\Delta dotA$ condition is not necessary to support the main message and the conclusions of this figure.

As mentioned above, we used H1 in order to have the staining of the control on the same membrane as the H3K14 modification, as it is more suitable for internal quantification of the WB signal.

Figure 2I: Since the $\Delta lphD$ strain is attenuated for replication, could differences be due to bacterial abundance?

As we measure the differences at 7h post infection, the difference cannot be due to different replication as the replication does merely start at this time point.

This is true, but differences in bacterial abundance can be mediated by differences in phagocytosis and it is unclear whether this is the case since all intracellular replication data are shown as fold changes in CFU normalized to 2h post-infection for each strain. The authors could consider presenting growth data as CFU/mL or showing CFU counts after 2h of infection.

Answer:

We presented the replication assays in THP1 cells with a normalization at 2h because this is what is usually asked for this type of experiments. However, we controlled that there are no differences in phagocytosis between the wild-type and the mutant strain. Please find below the growth curve for intracellular replication in THP1 cells without the normalization to T2:

Lines 205-209 . Experiments using the $\Delta lphD \Delta romA$ strain are needed to draw the conclusion that RomA is responsible for the methylation observed.

Nb: corresponds to fig 2J CHIP LphD

The reviewer might have overseen that we have published this result already in our Cell Host Microbe article Rolando. et al. 2013. There we have done CHIP with RomA and shown that it is responsible for H3K14 methylation (Rolando et al 2013). Thus, we cannot republish it here.

I agree that the previous report nicely shows that RomA methylates H3K14; however, data from a previous publication cannot serve as a control in the present study. Results from previous studies are often confirmed as controls in manuscripts, so I am confused about perceived issues 'republishing' the RomA phenotype. Without data showing that methylation is abolished by additional loss of *romA* from LphD-deficient bacteria, causation can merely be hypothesized and the text in lines 203-208 should be revised accordingly.

Answer:

As requested, we have changed the sentence as follows (line 205): "Methylation of H3K14 at these promoters, driven by RomA, is also helped by LphD, binding to the same regions of the chromatin (former Figure 3E now 4E)."

Figure 3A-D. Please perform statistical analyses at each time point and indicate at which time points growth differs significantly.

As requested by the reviewer we added the statistical analyses in the figures and figure legends.

Lines 903-905 it reads now... Non-linear regression analysis using a straight-line model showed a significant difference in the slopes of the two conditions (P=0.0291).

Figure 4A. There appears to be less LphD Y392F bound to RomA compared to the wild-type. Is this phenotype consistent across experiments?

A negative control should be included here. An unrelated HIS-tagged effector would be appropriate. Why is there a band - presumably corresponding to GST-MYL1 - in the LphD Y392F-GST-RomA pulldown? Please present the gels showing GST-fusions as one and include molecular weight markers.

As requested by the reviewer we repeated the experiment by adding an unrelated His-tagged effector that is known to be secreted by the T4SS. We choose His-LegK1 (Lpp1439) (Ge et al., 2009; PNAS 106(33):13725-30), as negative control. One can see now in **Figure 4A** that LphD binds RomA *in vitro* but LegK1.

Lines 282-284. Please show a representative Western blot in the supplement (like Fig S3F).cit” This revealed that inhibition of KAT7 activity abolishes the decrease in H3K14 methylation observed in infection with *L. pneumophila* lacking LphD (Δ *lphD*). Thus, LphD counteracts KAT7 activity (**Figure 4J**)

As requested, we included a representative western blot in **Figure S5H**

Lines 290-291 it reads now... Thus, LphD counteracts KAT7 activity (**Figure 4J** and **Figure S5H**)

All experiments where chemical inhibitors are used (e.g. Fig 4J) should include a vehicle control (e.g. DMSO)

As requested, we have done these experiments also by adding as vehicle control EtOH because the inhibitor was dissolved in EtOH. The results of this experiments added below show that the result is not dependent on EtOH.

Quantification of western blot signal for H3K14 methylation in THP1 cells treated 7 hours with either EtOH (0.01% v/v) or 1μM MWM-3835 (KAT7-specific inhibitor). THP-1 cells were treated 18 hours prior to infection with *L. pneumophila* wild type or Δ *lphD* strain expressing EGFP. Histones were isolated from FACS-sorted cells and analysed by western blot. Histone H1 was used as loading control and signal is compared to non-infected cells EtOH treated ($n = 3 \pm$ SEM).

Minor points:

Line 409: Should say '...predicted structural similarity...'

We corrected the title of the figure as requested.

Figure **1D** and **1F** should be removed since Figures 1G and 1H show the same thing and include genetic complementation.

We merged Figures 1E and 1F to have a unique **Figure 1E**. However, we did not remove it as reviewer#3 asked for having more understandable peptide predictions.

Ensure that the number of independent experiments are clearly indicated in each figure legend.

We apologize if this was not done well enough. We have now clearly indicated the number of independent experiments in each Figure

Why isn't Figure S2G a main figure? It can take the place of Figure 3A. Figure 3F can be moved to the supplemental information

As requested, we moved the THP1 complementation to the main figure, now **Figure 3C**, and moved the RNAseq data in the Supplemental, now **Figure S5A**.

Lines 310-312 cit" Indeed, a putative NLS was predicted *in silico* at the N-terminus of LphD; however, its deletion did not affect nuclear translocation of LphD (data not shown)" and 324-326. "This is seen in transfection when high amounts of protein are delivered into a host cell, as LphD transfection in HeLa cells leads to the deacetylation of several other residues such as H3K18 and H3K23 (data not shown)". Please present these data in the results section (can go in the supplement) or remove the statement.

As requested, we have added these data. The experiment where the predicted NLS was deleted is now shown in **Figure S6A**

Lines 317-319 it reads now... Indeed, a putative NLS was predicted *in silico* at the N-terminus of LphD; however, its deletion did not affect nuclear translocation of LphD (**Figure S6A**).

The results for the levels of H4K18ac and H3K23ac after transfection of LphD are now shown in **Figure S6B**

Lines 331-334 it reads now... This is seen in transfection when high amounts of protein are delivered into a host cell, as LphD transfection in HeLa cells leads to the deacetylation of several other residues such as H3K18 and H3K23 (**Figure S6B**).